# Development and Prediction of a Non-Destructive Quality Index (Qi) for Stored Date Fruits Using VIS–NIR Spectroscopy and Artificial Neural Networks

**DOI:** 10.3390/foods14173060

**Published:** 2025-08-29

**Authors:** Mahmoud G. Elamshity, Abdullah M. Alhamdan

**Affiliations:** Chair of Dates Industry and Technology, Department of Agricultural Engineering, College of Food and Agricultural Sciences, King Saud University, Riyadh 11451, Saudi Arabia; melamshity@ksu.edu.sa

**Keywords:** ANN, date fruits, modeling, microbial, physical and mechanical properties, quality index, storage, VIS-NIR

## Abstract

This study proposes a novel non-destructive approach to assessing and predicting the quality of stored date fruits using a composite quality index (Qi) modeled via visible–near-infrared (VIS–NIR) spectroscopy and artificial neural networks (ANNs). Two leading cultivars, Sukkary and Khlass, were stored for 12 months using three temperature regimes (25 °C, 5 °C, and −18 °C) and five types of packaging. The samples were grouped into six moisture content categories (4.36–36.70% d.b.), and key physicochemical traits, namely moisture, pH, hardness, total soluble solids (TSSs), density, color, and microbial load, were used to construct a normalized, dimensionless Qi. Spectral data (410–990 nm) were preprocessed using second-derivative transformation and modeled using partial least squares regression (PLSR) and the ANNs. The ANNs outperformed PLSR, achieving the correlation coefficient (R^2^) values of up to 0.944 (Sukkary) and 0.927 (Khlass), with corresponding root mean square error of prediction (RMSEP) values of 0.042 and 0.049, and the relative error of prediction (REP < 5%). The best quality retention was observed in the dates stored at −18 °C in pressed semi-rigid plastic containers (PSSPCs), with minimal microbial growth and superior sensory scores. The second-order Qi model showed a significantly better fit (*p* < 0.05, AIC-reduced) over that of linear alternatives, capturing the nonlinear degradation patterns during storage. The proposed system enables real-time, non-invasive quality monitoring and could support automated decision-making in postharvest management, packaging selection, and shelf-life prediction.

## 1. Introduction

Date palm (*Phoenix dactylifera* L.) is a cornerstone crop in arid and semi-arid regions due to its nutritional, economic, and cultural importance [1,2,3]. In 2024, global production reached 9.75 million tons, with Saudi Arabia contributing approximately 1.9 million tons from 31.5 million palms [4]. Dates mature through distinct stages Hababouk, Kimri, Khalal, Bisir, Rutab, and Tamr each with specific physicochemical characteristics [5]. At the Tamr stage, dates are consumed in soft to dry forms depending on the cultivar and moisture content [6].

Despite their importance, postharvest losses in date fruits can reach up to 26%, primarily due to suboptimal storage, insect infestation, or environmental exposure [7]. These losses affect both the quantitative yield and qualitative attributes such as texture, color, and nutritional value. The proper packaging and cold storage are crucial to mitigating such deterioration [8,9,10,11].

Non-destructive methods have become essential in postharvest quality monitoring, offering rapid and contactless evaluations of internal quality traits [12]. Among these, visible–near-infrared (VIS-NIR) spectroscopy has shown great promise for analyzing food matrices by capturing the interaction of light in the 400–1000 nm range with internal constituents, correlating strongly with moisture, sugar content, texture, and color [13,14].

The recent integration of VIS-NIR with machine learning, particularly artificial neural networks (ANNs), has enhanced its ability to model complex, nonlinear spectral relationships. ANNs have been used in agriculture, environmental science, and food processing to predict biological and physicochemical variables where the traditional linear methods fall short [15,16,17].

Previous research has applied VIS-NIR and ANN techniques to various food products, including tea, coconut oil, green coffee, and apples [18,19,20,21,22,23,24,25,26,27].

Although the applications of NIR have extended to fresh Barhi dates [28], fermented milk beverages containing dates [29], and flavored milk [30], limited research has quantitatively assessed dates at the Tamr stage using a comprehensive, unified index under real-world storage conditions. Furthermore, no prior study has integrated both objective (physicochemical) and subjective (sensory) quality attributes into a single, normalized quality measure.

To address this gap, this study proposes the development of a novel, non-destructive quality index (Qi) based on VIS–NIR spectral data. This Qi serves as a composite, dimensionless indicator integrating moisture content, pH, hardness, water activity, soluble solids, browning index, color difference, microbial load, and sensory scores. It facilitates a longitudinal assessment of fruit quality across varied moisture contents, packaging materials, and temperatures over a 12-month storage period. By incorporating both partial least squares regression (PLSR) and ANN modeling, Qi provides a practical and scalable tool for shelf-life prediction and quality assurance in date supply chains.

This study is therefore original in its holistic approach fusing spectral sensing, chemometric modeling, and ANN-based learning to non-destructively quantify and forecast the quality dynamics of date fruits in postharvest systems.

## 2. Materials, Methods, and Measurements

### 2.1. Materials

#### 2.1.1. Dates

Date fruits from two commonly produced varieties in Saudi Arabia, Sukkary and Khlass, were cultivated. They were harvested at the Tamr stage, characterized by their brown to black color [31,32]. The Sukkary and Khlass dates were obtained from the King Saud University Farm Station in the Derab region (80 km southwest of Riyadh, Saudi Arabia). Figure 1 shows samples of the palm trees used in this study during cultivation.

The dates were harvested at the Tamr stage from 10 palm trees from each cultivar and placed into wrapped polyethylene bags packed into cardboard boxes (20 kg). Then, the fruits were transported to the Food Engineering laboratory at the College of Food and Agriculture Sciences, KSU, in Riyadh. They were cleaned using a directed jet of compressed air to remove surface dust and loose debris before sorting. This cleaning method was chosen to minimize physical handling and preserve the integrity of the fruits’ surfaces for subsequent measurements. Then, the fruits were sorted into intact and consumable fruits suitable for analysis, while damaged fruits exhibiting physical defects such as cracks, bruises, or pest marks and non-consumable fruits showing signs of microbial spoilage or advanced desiccation were excluded. Samples of the ready-to-measure fruits are shown in Figure 2.

Six moisture groups were chosen to simulate the moisture content levels of commercial date fruits at the Tamr stage. It was not feasible to obtain enough dates during harvesting at a uniform MC due to the large variations in the MC among dates on the same cluster at any harvesting time. There is a need to have consistent groups of dates within a wider range of MCs for NIR analysis [33]. A controlled hydration protocol was employed by incrementally adding pre-weighed amounts of distilled water to harvested low-moisture dates (3.562 to 7.273% db.). The raw dates were carefully moisturized using six dosages of distilled water to meet the designated MC ranges shown in Table 1. Initially, a calculated weight of water was gently sprayed onto the samples, which were then sealed into containers and stored at 5 °C for 72 h to ensure uniform moisture absorption throughout the date’s matrix. To further ensure MC equilibrium was reached in the dates, the samples were placed into five controlled relative humidity chambers and allowed to equilibrate. Moisture equilibration was confirmed by monitoring the sample weights until stabilization, indicating a uniform distribution of moisture within the dates’ tissues. The obtained dates could be utilized for sampling into six groups based on their moisture content levels, as shown in Table 1. This approach is commonly used in food quality research to standardize the moisture content across sample groups for comparative analysis [34,35,36]. It enables researchers to isolate the effect of moisture content from other variables, which is crucial for developing predictive models, such as the quality index (Qi) used in this study. Furthermore, previous studies on fruit drying, rehydration, and water activity modeling have adopted similar moisture adjustment techniques to mimic physiological moisture variations [37,38]. To validate the uniformity of the moisture distribution, random subsamples from each group were taken post-equilibration and tested for their moisture content using a standard oven-drying method [39]. The low standard deviation observed (<0.5%) confirmed satisfactory moisture homogeneity within each treatment group.

All examined samples were scanned using the Felix 750, a handheld VIS-NIR spectrometer (Model F-750, Firmware v1.2.0 build 7041, Felix Instruments, Camas, WA, USA). The physicochemical properties of the six groups of dates were then measured, and sensory evaluations were made. The data was used for spectral data acquisition and analysis in the NIR assessment.

#### 2.1.2. Packaging Materials

Five commercial packaging formats were used for date storage: (1) open cardboard cartons (OCCs), (2) covered cardboard cartons (CCCs), (3) covered cardboard cartons with a sealable plastic bag (CCSPBs), (4) semi-rigid plastic containers (SSPCs), and (5) pressed semi-rigid plastic containers (PSSPCs). The barrier performance is summarized in Table 2, reporting the water vapor transmission rate (WVTR; g·m^−2^·day^−1^) measured at 38 °C/90% RH (ASTM F1249) and the oxygen transmission rate (OTR; cm^3^·m^−2^·day^−1^·bar^−1^) measured at 23 °C/0% RH (ASTM D3985/ISO 15105-2) [40,41,42,43,44,45]. Where applicable, the light transmittance/opacity for the plastic formats is provided per ASTM D1003 (paperboard is opaque and was not assessed).

The material constructions were as follows OCC/CCC = corrugated Kraft paperboard; CCSPB = a paperboard carton plus a heat-sealable food-contact poly bag; and SSPC/PSSPC = food-contact rigid/semi-rigid polymers—as specified on the suppliers’ technical data sheets. All the packages had a nominal 3 kg capacity; three package replicates were prepared per treatment. Note that the paperboard barrier values vary strongly with relative humidity, and the plastic barrier values scale with thickness; multilayer structures containing EVOH (PSSPCs) serve as the high-barrier benchmark under dry conditions (typical OTR < 1) [50,51,52,53,54,55].

The packaging formats were selected following field surveys and consultations with local producers and vendors in Saudi Arabia. Representative suppliers included cartons/paperboard from Obeikan Packaging, Easternpak (INDEVCO), Gulf Carton Factory, and Riyadh Carton; flexible bags (LDPE/LLDPE) from Napco National, Al Watania Plastics, and Takween Advanced Industries (and subsidiaries); and rigid/semi-rigid container (PET/PP/EVOH) from Obeikan Solutions, Al Watania Plastics, and Takween (Plastico SPS). The suppliers’ names are provided for traceability and do not imply endorsement.

All materials were procured from certified food-grade suppliers in Saudi Arabia and verified for compliance with FDA CFR Title 21 and EU Regulation No. 1935/2004 (and EU 10/2011 for plastics) [46] via the suppliers’ Certificates of Conformance/food-contact Declarations. Each package bore the internationally recognized food-contact [56,57] SSPCs and PSSPCs, though less common on the date market, were intentionally included for their enhanced sealability, rigidity, and transparency, which are advantageous for moisture and microbial control.

### 2.2. Methods

#### Storage of the Dates

After the initial measurements of the dates, they were put into specified packages (3 kg each). The fruits were then stored at three temperatures (25, 5, and −18 °C).

Storage at 25 °C, representing ambient or room-temperature conditions, is commonly employed in postharvest studies as a control treatment, especially in top date-producing countries such as Egypt, Saudi Arabia, and Iran. This temperature reflects typical market and household storage environments where refrigeration is limited. Assessing fruit quality under 25 °C conditions is valuable for understanding the rate and extent of deterioration in traditional storage systems, which are still widely used in local retail and supply chains [58]. Storage at 5 °C corresponds to the conventional refrigeration conditions commonly used to extend the shelf life of fresh and semi-dried fruits and is less expensive compared to freezing storage. At this cold temperature, enzymatic activity and microbial growth are substantially reduced, thereby minimizing spoilage. Refrigeration at 5 °C is effective in maintaining fruit’s texture, flavor, and nutritional content and is standard practice in commercial cold-chain logistics and domestic storage [58].

The selection of −18 °C for storing dates is based on the superiority quality and global practices for frozen food, as outlined by the Codex Alimentarius and endorsed by both the Food and Agriculture Organization [4,59] and the U.S. Food and Drug Administration (FDA) [60]. At −18 °C, microbial and enzymatic activities are effectively inhibited, and oxidative reactions are greatly reduced. Previous studies have demonstrated that −18 °C storage effectively maintains the physicochemical properties, sensory quality, and nutritional value of dates and other low-moisture fruits over extended periods [28]. Accordingly, the inclusion of −18 °C in this study reflects practical industrial and export-oriented storage practices for dates [29]. Measurements of the physicochemical properties of the samples were conducted every month for 12 months. Figure 3 illustrates the preparation, packaging, and storage of the date fruits and measurement of various microbial, sensory, and physicochemical properties of these fruits.

The storage environment (RH and light): At 25 °C (in the ambient room), the average relative humidity was maintained at ~50 ± 5% RH, and at 5 °C (in the cold room), the average RH was held at ~70 ± 10% RH, with the RH in both cases continuously monitored using a calibrated data logger. For the −18 °C freezer, RH is not a stable or meaningful control parameter because of the very low absolute humidity and frost control cycles; accordingly, we report the temperature stability (−18 ± 1 °C) and minimal door-open time instead. Light exposure was controlled by storing all the samples in the dark (lights off; <1 lux). For the measurements, the samples were removed once, transported in covered containers, equilibrated as needed (to prevent condensation for chilled/frozen lots), measured, and not returned to storage. The VIS–NIR acquisitions were performed using a black sampling hood to exclude ambient light.

### 2.3. Measurements

The physicochemical properties (mass (m), dimensions (L, D), moisture content (M.C.), water activity (aw), total soluble solids (TSSs), browning index (BI), total color difference (∆E), hydrogen potential (pH), and hardness (h)) of the stored date fruits (SDFs) were measured. There was a total of 105 replicates for each of the six sample groups, with 3 levels of storage temperature, 5 types of packaging materials, and 7 repetitions for each (3 × 5 × 7 = 105), as illustrated in Figure 3.

(a)Physical properties

A balance (model MA3002, Mettler, Switzerland-3200 g, ±0.01 g) was used to measure the mass of the date fruits. The dimensions of the fruits (diameter and length) were measured using a digital vernier caliper (CD-15CW, Mitutoyo Corporation, Kanagawa, Japan).

The moisture content was measured utilizing a vacuum oven (Vacutherm model VT 6025, Heraeus Instrument, D-63450, Hannover, Germany) at 70 °C under a 200 mmHg vacuum for 48 h [61]. A water activity meter (Series 4TE, 4TEV, DUO Version 4, Decagon Devices, Inc.) was used to measure the water activity at room temperature (25 °C). The total soluble solid (TSS) contents of the dates were measured using a refractometer (MAN96801 09/14, Hanna Instruments Inc., Woonsocket, RI, USA). The pH of the dates was measured using a pH meter (model Five Go ™ F2 pH/mV Meter, Bibby Scientific Ltd., Mettler Toledo, GmbH, Greifensee, Switzerland) with a measuring accuracy ± 0.01 and standardized with a pH 7.0 buffer.

The basic color coefficients (L*, a*, and b*) of the dates were measured using a color instrument (Color 45/0, Hunter Associates Laboratory, Inc., Reston, VA, USA). L* indicates whiteness (brightness)/darkness, a* indicates redness/greenness, and b* indicates yellowness/blueness.

The color derivatives of chroma, hue angle, browning index (BI), and total color difference (∆E) were calculated from basic color coefficients [30,38,62,63,64] according to(1)$$Chroma={\left({a^{*}}^{2}+{b^{*}}^{2}\right)}^{0.5}$$(2)$$Hueangel={tan}^{-1}\left(\frac{b^{*}}{a^{*}}\right)$$(3)$$BI=\frac{\left[100-\left(x-0.31\right)\right]}{0.17}$$
where $x=\frac{\left(a^{*}+1.75\times l^{*}\right)}{\left(5.645\times l^{*}+a^{*}-3.012\times b^{*}\right)}$(4)$$\Delta E={\left[(L_{0}^{*}-L^{*}{)}^{2}+(a_{0}^{*}-a^{*}{)}^{2}-(b_{0}^{*}-b^{*}{)}^{2}\right]}^{0.5}$$

(b)Mechanical properties

A texture analyzer (TA-HDi, Model HD3128, Stable Micro Systems, Surrey, England) was utilized for the TPA test. The fruit samples (with pits) were tested with a cylinder (P-75) and the rod velocity set to 1.5 mm/s, reaching a depth of 5 mm deformation. This compression (bite) was repeated twice to generate texture profile curves. The resulting force–time deformation curves provided several parameters, including hardness (indicating the maximum force needed to compress the sample to the specified fruit height), adhesiveness, fracturability, cohesiveness, resilience, chewiness, gumminess, and springiness [65,66,67,68,69,70,71,72]. The data was analyzed using Texture Expert Exceed software (version 2.05, Stable Micro Systems) and Excel to calculate the hardness (N) of the fruit.

(c)Chemical analysis

All chemical analyses of the date fruits were conducted according to the standards of the Association of Official Analytical Chemists [61]. These include (i) moisture content determination via oven-drying (AOAC 934.06); (ii) total soluble solid (TSS) measurement using a digital refractometer (expressed in °Brix); and (iii) pH assessments using a calibrated bench-top pH meter. All the procedures followed standardized, validated protocols to ensure reproducibility and scientific rigor.

(d)Assessment of Microbial Quality

A standard procedure [61] was implemented to evaluate the total viable count (TVC) and yeast enumeration (YE) for the fresh and stored dates. Briefly, 25 fruits from each package were seed pitted. The fruit’s flesh (25 g) was added to 225 mL of sterilized saline solution (0.85%) and homogenized using a Stomacher^®^ 400 circulator (Seward GmbH, West Sussex, UK). The homogenate was diluted (10-fold) with sterile saline solution until the optimal dilution was reached. Then, 1 mL of the diluted homogenate was pour-plated onto CM0309 nutritional agar media (Oxoid, Basingstoke, Hampshire, UK) for the TVC analysis, whereas potato dextrose agar was used for YE cultivation. Incubation was conducted at 37 °C for 24–48 h for the TVC enumeration, whereas it proceeded at 25 °C for 2–5 days for YE enumeration. The counts were expressed as log CFU/g based on triplicate samples. The methods for aerobic plate counting and for yeast/mold, plate media, the incubation conditions, and the detection limits are specified in ISO 4833-1 [73] and ISO 21527-2 [74], respectively.

(e)Sensory evaluation

A group of sixty-six trained evaluators from King Saud University at the College of Food and Agricultural Sciences conducted sensory evaluations on the fresh and stored date fruits. Safety and health precautions were observed in conducting the planned sensory assessments. The evaluation process utilized a 9-point Hedonic scale used by various researchers [30,62,75,76,77,78,79,80,81]. These sensing tests were based on texture, taste, hardness, color, adhesion, peel ability, and overall acceptability. The responses ranged from 1 (indicating “extremely dislike”) to 9 (indicating “extremely like”).

The sensory sample preparation and serving protocol: The dates were not washed with water when served to avoid altering their surface hydration; instead, the fruits were dry-cleaned at intake (with compressed air) and surface-wiped with sterile, food-grade wipes using sanitized stainless tools and powder-free gloves. Immediately before serving, the fruits were manually pitted and cut into uniform pieces (≈2 × 2 cm; ~10 g per portion). Each panelist received one 10 g portion per treatment prepared from a composite of three fruits from the same lot. The sensory tests were performed in individual sensory booths equipped with neutral lighting (D65), a controlled temperature (22 ± 1 °C), and adequate ventilation, following the ISO 8589 [82], guidelines for sensory analysis environments. The samples were assessed at room temperature for ~30 min (to prevent condensation in chilled/frozen lots) and presented in opaque, odor-free cups labeled with randomized three-digit blinding codes. The presentation order was randomized and counterbalanced (with a Williams Latin square) [77,83,84,85] to control position and carry-over effects; the inter-sample intervals were ≥60–90 s, using room-temperature water and unsalted crackers as palate cleansers. Each lot was served once and not returned to storage. The sensory evaluation took place quarterly (at 3, 6, 9, and 12 months) throughout the storage period.

When releasing date samples to a trained sensory panel, the safest approach is to use conservative acceptance (action) limits informed by institutional food safety SOPs based on the ICMSF/Codex principles and ISO methods [60,82,86,87,88,89,90,91,92,93,94,95,96,97,98,99]. Because these limits are product-based, not temperature-based, we applied the same criteria to the lots whether they had been stored at −18 °C, 5 °C, or 25 °C, provided they had been handled under hygienic conditions. Table 3 presents a concise set of panel-release criteria suitable for low-moisture, ready-to-eat dates (at the Tamr stage), with the analytical methods typically used:

For each sensory session, a composite sample from the candidate batch was plated in duplicate for the TVC and YE assessments. Any result above the Reject threshold led to immediate exclusion of the batch from service. Results within the Investigate/Hold band triggered re-plating and a review of the handling records; only lots that subsequently complied with the acceptance limits were released to the panel.

Although pathogens were not targeted in this study, every unit was visually inspected for mold; any lot showing visible colonies was discarded. The release criteria were applied uniformly regardless of storage temperature (−18 °C, 5 °C, or 25 °C) and were evaluated on the day of service.

### 2.4. Modeling

(a)Quality Index (Qi) Prediction

The quality index (Qi, ranging from 0 to 1) serves as a tool for standardizing and representing the variables relative to the minimum value of the controlled variable and is used by several researchers [100,101,102]. To standardize the parameters, the following formula can be applied:(5)Xi^=Xi−XminXmax−XminXi^ represents the normalized value of the quality parameter x, while $x_{i}$ stands for the value of the quality parameter measured. $x_{max}$ and $x_{min}$ denote the maximum and minimum values of the quality parameter x across the entire dataset, respectively. The quality index $Q_{i}$ was computed as follows:(6)Qi=∑i=1NXi^N
with ‘N’ representing the number of samples. The generated $Q_{i}$ will accommodate both normalized objective sample properties and the overall sensory data. Furthermore, the quality index data undergoes further assessment using NIR spectroscopy to predict Qi and food properties non-destructively.

(b)The VIS-NIR Technique

Before measuring the quality parameters of the fresh dates, each fruit was scanned using the F-750’s optical lens. In this work, 15,000 scans were recorded with a handheld NIR meter (F-750, Firmware v.1.2.0 build 7041, Felix Instruments, Camas, WA, USA) to model and predict the date fruits’ properties and other quality parameters. The F-750 device was equipped with a reference shutter, allowing for the calculation of dark current and ambient light effects with each measurement, particularly when scanning with the lamp turned off.

The F-750 device integrates a Zeiss MMS1 VIS–NIR spectrometer, featuring a full spectral operating range of 285–1200 nm, which captures data at 3 nm intervals. While this range encompasses the ultraviolet (UV), visible (VIS), and near-infrared (NIR) regions, a 410–990 nm segment of UV–VIS–NIR was selected. This VIS–NIR range was chosen to contain the most informative and consistent spectral signatures for modeling the physicochemical properties of the date fruits. Reflectance measurements involve illuminating a sample with light and quantifying the light reflected from it. The reflected light comprises specular and diffuse components, collectively known as the total reflected light [103,104,105]. The light directed onto the sample is termed incident light, while the angle formed between this incident light and the sample is known as the angle of incidence. Relative reflectance measurements determine the proportion of light reflected from a sample’s surface relative to that from a reference plate. This calculation assumes that the reference plate reflects 100% of the incident light [106,107]. Reflectance spectroscopy, characterized by point-probe technology, offers several advantages. It easily fits through the accessory channel of a standard diagnostic endoscope, and its fiber optics maintain a consistent geometry between the light source and the detector, ensuring predictable measurements [108]. The measurements in this study were conducted using reflective spectroscopy mode, where near-infrared electromagnetic radiation was directed onto the date fruit’s surface and the diffusely reflected light was captured by an integrated sensor. The device includes a halogen tungsten light source, a Zeiss MMS1 VIS-NIR spectrometer equipped with a diffraction grating dispersive element, and a silicon-based photodiode array detector.

To ensure accurate spectral acquisition, each fruit sample was scanned in triplicate by placing it in direct contact with the device’s optical window (equipped with a sapphire lens), which maintained a fixed geometry and minimized stray light. The device’s black rubber sampling hood provided shielding from ambient light. No transmission or absorbance modes were used, as the method relied entirely on diffuse reflectance measurements. Preprocessing NIR spectrum data is crucial to developing robust models with a high performance. Utilizing numerical derivatives can be particularly beneficial in NIR spectroscopy in eliminating extraneous signals from the spectra [109,110].

Before the capture process, care was taken to ensure complete contact with the date’s surface with the lens of the F-750. Variability in the surface texture and shape of the fruits introduced inconsistency into the light reflection, impacting the spectral accuracy. A specialized holder was designed to stabilize the fruit samples and maintain consistent contact between the sample surface and the light source. Furthermore, signal preprocessing techniques, such as multiplicative scatter correction (MSC), were applied to minimizing noise from surface irregularities [111,112,113]. After recording the spectrum, the data was transferred from the F-750 to a PC for analysis. The data was stored in comma-separated values file (CSV files) using Data Viewer software (Version 2.1.7).

In this study, the Savitsky–Golay (SG) algorithm was applied specifically to computing the second derivative of the spectral data, not solely for smoothing purposes. This second-derivative transformation serves two main objectives: (1) enhancement of subtle spectral features by resolving overlapping peaks and minimizing the baseline drift, thereby improving the signal-to-noise ratio, and (2) improvement of the model’s interpretability and robustness during the chemometric analysis, particularly when employing partial least squares regression (PLSR) and artificial neural networks (ANNs), as it accentuates the informative spectral regions associated with specific physicochemical attributes.

To develop and evaluate the calibration models, a total of 15,000 spectral samples were collected. These were systematically divided into three subsets: 10,500 spectra samples (70%) were used for model training, 3000 spectra samples (20%) were used for testing, and 1500 spectra samples (10%) were used for independent validation. The splitting procedure ensured that spectra from the same instrumental replicates were not simultaneously present in multiple subsets to prevent data leakage and preserve model independence. This approach ensures that the evaluation metrics accurately reflect the model’s generalization ability rather than memorizing repeated patterns. Two analysis tools, partial least squares regression (PLSR) and artificial neural networks (ANNs), were utilized to develop calibration models.

The artificial neural network (ANN) model was trained using preprocessed VIS–NIR spectral data (410–990 nm) collected from individual date fruit samples. The model’s output was a dimensionless, continuous quality index (Qi) derived from the integration of the normalized physicochemical and sensory parameters. This nonlinear regression approach allowed the ANN to establish complex relationships between the spectral patterns and overall fruit quality, thereby enabling accurate, non-destructive predictions of the quality dynamics throughout storage. In partial least squares regression (PLSR), the selection of the latent variables (LVs) is driven by the goal of maximizing the shared variance between the predictor matrix (X), which in this case comprises the VIS–NIR spectral data (410–990 nm), and the response variable (Y), represented by the reference quality index (Qi) [114]. PLSR achieves this by sequentially extracting orthogonal components—known as latent variables—that capture the directions in X most relevant for predicting Y, thereby reducing the dimensionality while retaining a meaningful covariance structure [115]. In the context of this study, the LVs were computed to uncover and model key spectral signatures associated with the physicochemical and sensory attributes of the date fruits. For instance, the first latent variable (LV1) predominantly represented the spectral variance linked to moisture content and surface color, both of which strongly influence light reflectance. The second latent variable (LV2) was more reflective of variations in firmness and soluble solids, corresponding to the specific absorption behaviors in the near-infrared region. These extracted LVs enabled the PLSR model to translate complex, high-dimensional spectral data into a robust predictive framework for estimating date fruit quality in a non-destructive manner.

Simultaneously, artificial neural networks (ANNs) were utilized, offering a robust nonlinear pattern recognition method capable of modeling complex diversities, environmental influences, and instrument fluctuations [116]. The analyzed data was processed using AppBuilder v2.1.7 software (Felix Instruments, Camas, WA, USA) [28,29,30].

In chemometric modeling, particularly in the context of VIS–NIR spectral analyses, evaluating the model’s performance requires robust statistical metrics that reflect both the goodness-of-fit and predictive reliability. Three widely accepted indicators include the correlation coefficient (R^2^), the root mean square error in calibration (RMSEC), and cross-validation (RMSECV) [14,15,30,117,118]. These metrics collectively help quantify the model’s ability to explain the variance in the target variable and its capacity to generalize to unseen data.

The correlation coefficient (R^2^) measures the proportion of the variance in the reference (measured) values that are explained by the model and is calculated as(7)$$R^{2}=1-\frac{\sum_{i=1}^{n}{\left(y_{i}-\hat{y_{i}}\right)}^{2}}{\sum_{i=1}^{n}{\left(y_{i}-\bar{y}\right)}^{2}}$$
where $y_{i}$ is the observed value, $\hat{y_{i}}$ is the predicted value, $\bar{y}\;$ is, the mean of the observed values, and *n* is the number of observations.

The RMSEC quantifies the average deviation between the predicted and observed values during the calibration phase:(8)$$RMSEC=\sqrt{\frac{1}{n}\sum_{i=1}^{n}{\left(y_{i}-\hat{y_{i}}\right)}^{2}}$$

Similarly, the RMSECV is derived from cross-validation, where subsets of the calibration data are iteratively left out and predicted by the model trained on the remaining data. It provides a more realistic estimate of the model’s future performance:(9)$$RMSECV=\sqrt{\frac{1}{n}\sum_{i=1}^{n}{\left(y_{i}-\hat{y_{cv,i}}\right)}^{2}}$$
where $\hat{y_{cv,i}}$ denotes the cross-validated prediction of $y_{i}$.

Together, these metrics enable a comprehensive evaluation of the model’s accuracy, overfitting risk, and robustness, crucial aspects when building predictive tools for food quality assessments using spectral data.

In addition to conventional performance metrics such as the R^2^, RMSEC, and RMSECV, this study also employs the Relative Error of Prediction (REP), Range Error Ratio (RER), and Root Mean Square Error of Prediction (RMSEP) values [14,117,119] to provide a more nuanced evaluation of the model’s robustness and predictive accuracy. These metrics are particularly valuable in assessing the generalizability across varied sample conditions and enhancing the interpretability when comparing across models.

The Relative Error of Prediction (REP) expresses the prediction error relative to the meaning of the measured values and is calculated as(10)$$REP\left(\%\right)=\left(\frac{RMSEP}{\bar{y}}\right)\times 100$$
where RMSEP is the Root Mean Square Error of Prediction, and $\bar{y}$ is the meaning of the reference (observed) values in the validation set.

The Range Error Ratio (RER) is defined as the ratio of the range of reference values to the RMSEP, quantifying the model’s ability to discriminate across the entire range of variation:(11)$$RER=\left(\frac{y_{max}-y_{min}}{RMSEP}\right)$$

These metrics complement traditional R^2^ values by contextualizing the magnitude of the prediction error concerning the data’s spread and central tendency. For validation, a separate test set comprising 1500 independent samples was used to compute the RMSEP as(12)$$RMSEP=\sqrt{\frac{1}{n}\sum_{i=1}^{n}{\left(y_{i}-\hat{y_{i}}\right)}^{2}}$$
where $y_{i}$ is the observed reference value, $\hat{y_{i}}\;$ is the predicted value is, and *n* is the number of validation samples.

The inclusion of the REP and RER alongside the RMSEP allows for a more comprehensive and statistically meaningful interpretation of the model’s performance, particularly in non-destructive spectral assessments of fruit quality.

The ANN was trained on a large dataset of spectral readings, enabling it to account for noise and variability in the data. To ensure scientific rigor and reproducibility, the methodology includes detailed descriptions of the model’s development, training procedures, cross-validation strategies, and statistical performance indicators (RMSE, R^2^, and prediction bias). Comprehensive data preprocessing, comprising spectral smoothing, normalization, and outlier detection, was also systematically documented, enhancing the transparency and traceability of the analytical process.

### 2.5. Statistical Analysis

The statistical analysis was performed for quantifiable properties of the date fruits using statistical software (SAS software, Version 9.4, Rev. 940_23w05, SAS Institute Inc., Cary, NC, USA) [102,120]. The experimental data, expressed as the means ± standard deviations (SD), were analyzed using a one-way analysis of variance (ANOVA) to assess statistically significant differences among treatment groups. The least significant difference (LSD) test was employed to identify pairwise differences at a significance level of *p* ≤ 0.05.

The prediction performance was evaluated through calibration and validation results using AppBuilder v2.1.7 software (Felix Instruments, Camas, WA, USA). Experimental data were collected utilizing the Microsoft Office 365 package (Microsoft, Redmond, WA, USA) [30]. Graphs, plots, and other computations were executed utilizing Python version 3.12.4.

All the data visualizations, including graphs, bar plots, and regression charts, were developed using the Python programming language (version 3.12.4) with analytical libraries such as SciPy and Matplotlib. This version corresponds to the bugfix maintenance release under PEP 693 by the Python Software Foundation (2001–2025) [121,122,123,124,125].

Initial data tabulation and management tasks were carried out using Microsoft Excel (Microsoft Office 365; Microsoft Corp., Redmond, WA, USA) [30,126].

## 3. Results and Discussions

The results provide a detailed analysis of the effect of the storage conditions on the physicochemical properties (monthly) and sensory and microbial evaluations (every 3 months) of the date fruits. The Qi and VIS-NIR were utilized to model the quality parameters, including both objective and subjective assessments of the stored date fruits, with a highly acceptable performance.

### 3.1. Modeling of the Quality Index (Qi)

Following the assessment of their sensory attributes and physicochemical properties over the storage period, a composite quality index (Qi) was calculated to represent the overall quality of the stored date samples quantitatively. The Qi incorporated 14 individual parameters, with 7 related to sensory evaluation and 7 to physicochemical characteristics, providing an integrated measure of the product quality throughout storage. A Qi value of 1.0 denotes the maximum quality, while values approaching 0.0 reflect progressive deterioration.

As shown in Figure 4, a gradual and consistent decline in the Qi values was observed across all storage durations, indicating a progressive deterioration in both the sensory and physicochemical attributes. The initial Qi values modeled were 0.988 for Sukkary dates and 0.984 for Khlass dates. By the end of the storage period, these values decreased to 0.713 and 0.665, respectively. These findings align with previous studies on date fruit quality degradation under storage conditions. For instance, Al-Habsi reported that Khlass dates stored in both ambient and refrigerated conditions exhibited marked declines in their flavor, texture, and total soluble solids over 6 months, suggesting a direct link between storage duration and reduced sensory acceptability [127]. Similarly, Mohamed observed significant decreases in the moisture content, sugar concentration, and color parameters in Barhi dates during postharvest cold storage, which correlated with diminished sensory scores [128]. These results reinforce the robustness of the Qi model in this study as a reliable integrative tool for capturing multi-attribute quality deterioration in dates stored.

A strong correlation between the Qi and the estimated shelf life was observed, with a high correlation coefficient (R^2^ > 0.983), as shown in Table 4. Previous studies have proposed that polynomial models effectively describe and predict the relationship between sensory attributes and physicochemical properties in food quality assessments [129,130]. Garcia and Patel investigated how polynomial models can be utilized to predict sensory responses based on objective indicators, including color, texture, and chemical composition [83]. Their findings support the use of such models to bridge subjective sensory evaluations with measurable quality traits. In alignment with these approaches, the Qi model in this study successfully captured the integrated effects of storage on both the sensory and physicochemical dimensions of quality, providing a robust indicator for monitoring postharvest changes in stored dates.

To objectively assess whether the second order (quadratic) regression model offered a statistically meaningful improvement over a simpler linear model, we conducted a model comparison using both the Akaike Information Criterion (AIC) and a one-way analysis of variance (ANOVA). As shown in Table 5, the quadratic models consistently produced lower AIC values for both the Sukkary and Khlass cultivars, indicating a more favorable trade-off between the model’s fit and complexity. Additionally, the ANOVA results revealed that the improvements in the model’s fit were statistically significant (*p* < 0.05), supporting the use of the quadratic model over the linear alternative [131,132].

These results suggest that the second-order model more effectively captures the subtle nonlinear trends in the degradation of the quality of date fruits over extended storage durations, particularly during the mid- to late storage stages when the physicochemical changes may not follow a linear path. This approach is consistent with established modeling strategies in food science, where polynomial functions are used to describe complex quality dynamics in biological matrices [131,132,133,134,135,136,137].

The quality index (Qi) demonstrated strong reliability and effectiveness as a comprehensive indicator for evaluating the overall quality of date fruits during a 12-month storage period under various packaging and temperature conditions. Its sensitivity to changes in both the sensory and physicochemical attributes makes it a valuable tool for tracking quality degradation over time. Furthermore, Qi has potential utility beyond research applications; regulatory authorities and governmental bodies could adopt this index to assess and monitor the actual quality and estimated shelf life of food products throughout the supply chain, including the production, storage, distribution, and retail stages.

The successful integration of sensory evaluations and physicochemical measurements into a single composite index supports its broader applicability in postharvest quality monitoring. Notably, this approach may be extended to other fruits and vegetables, offering a standardized and quantifiable method for assessing and potentially predicting shelf life in similar postharvest systems, provided the appropriate model training and validation are conducted for each commodity. Additionally, the modeling and prediction of Qi using non-destructive, rapid analytical techniques, such as near-infrared (NIR) spectroscopy, presents a promising direction for future research and industry adoption, enabling real-time quality monitoring with minimal sample preparation.

### 3.2. The VIS-NIR Spectral Analysis

In this study, the mean second derivative of the absorbance spectra was produced for 15,000 samples of the stored dates fruits (SDFs) utilizing a key spectral preprocessing technique to enhance the analytical resolution and model performance. This transformation offered several advantages over a raw spectral analysis, especially in the context of complex biological matrices such as date fruits, where overlapping absorption bands from water, sugars, and pigments complicate direct interpretation. First, the second derivative improves the peak resolution by mathematically accentuating subtle inflection points, thereby facilitating the discrimination of closely spaced spectral features. This is crucial for accurately capturing spectral signatures linked to quality attributes. Second, it serves as a robust method for baseline correction, effectively minimizing the signal drift and offset introduced by surface irregularities or scattering effects, thereby standardizing the spectra across varying storage conditions and packaging environments. Third, while first-derivative spectra highlight the direction and slope of changes in absorbance, the second derivative enhances the specificity by pinpointing the locations of the spectral band centers. This is particularly beneficial for selecting informative wave lengths in subsequent chemometric modeling. Finally, to address the inherent sensitivity of the derivative methods to noise, the Savitsky–Golay smoothing algorithm (a second-order polynomial with an optimized window size) was applied. This approach preserved essential spectral information while suppressing high-frequency noise, ensuring spectral integrity and interpretability.

Collectively, the second derivative of the VIS–NIR spectra (410–990 nm) improved spectral clarity and strengthened the predictive reliability of the models developed to assess the key physicochemical properties investigated, which were moisture content (M.C.), water activity (aw), total soluble solids (TSSs), browning index (BI), color difference (ΔE), pH, and hardness. Figure 5 shows an image of the raw spectral absorbance of the 15,000 samples of Sukkary and Khlass fruits under different storage conditions, utilizing the Flex F750 meter and Data Viewer software (Version 2.1.7). The initial spectral data acquired using the F-750 VIS–NIR spectrometer were obtained in reflectance mode, leveraging the device’s diffuse reflectance configuration. This setup measures the intensity of the light reflected from the surface of the fruit relative to that for a calibrated white reference, with the results expressed as a percentage of incident radiation. This approach enables the non-destructive acquisition of surface-level optical properties critical for quality assessments.

Figure 6 and Figure 7 show the reflectance spectra (%) measured for the storage durations of the Sukkary and Khlass fruits, respectively. In this study, the second derivative of the NIR spectra was employed. The second absorbance derivative of the NIR spectra was incorporated to improve the data quality. The mean second absorbance derivative spectra were created for the stored date fruits, comprising 3000 samples across the 327–1101 nm wavelength (λ) range. The recorded spectral reflectance for the date samples predominantly ranged from 393 to 1120 nm, aligning with the previously reported spectral windows used in agricultural and food quality analyses. This wavelength (λ) range is consistent with the findings of Jones and Smith, who identified similar spectral intervals as optimal for reflectance measurements in food matrices due to their sensitivity to compositional attributes such as moisture and sugars [104]. Moreover, Jiang demonstrated the applicability of the VIS-NIR spectrum (400–1100 nm) to detecting subtle chemical variations in biological and soil materials, even under varying moisture conditions [138]. When compared to these studies, the current reflectance profile for the date fruits further confirms the reliability of this spectral window for evaluating physicochemical attributes, notably those related to moisture content, sugar levels, and textural qualities, which are critical in assessing postharvest quality. Thus, the spectral behavior observed in our results reinforces the broader utility of the VIS-NIR region for non-destructive quality monitoring of date fruits and other similar natural products.

Figure 8 and Figure 9 show the second-derivative absorbance for the Sukkary and Khlass dates. Each color of the curves represents an average of 1000 fruit samples stored for each cultivar at a single temperature. Complete, detailed data from the visible–near-infrared spectroscopy results for the Sukkary and Khlass dates are available upon request.

The Sukkary dates from moisture content groups A through E were stored at −18 °C using pressed semi-rigid plastic containers and are labeled accordingly as Suk-A-F18-PSSPC, Suk-B-F18-PSSPC, Suk-C-F18-PSSPC, Suk-D-F18-PSSPC, and Suk-E-F18-PSSPC, respectively.

The Khlass dates from moisture content groups A through E were stored at −18 °C using pressed semi-rigid plastic containers and are labeled accordingly as KHL-A-F18-PSSPC, KHL-B-F18-PSSPC, KHL-C-F18-PSSPC, KHL-D-F18-PSSPC, and KHL-E-F18-PSSPC, respectively.

### 3.3. Quality Index (Qi) Modeling Correlated with the VIS-NIR Spectra

A further analysis of the VIS-NIR spectra was conducted using two robust techniques to analyze the correlation of the quality of the date fruits with the Qi. These two analytical methods were partial least squares regression (PLSR) and artificial neural networks (ANNs).

#### 3.3.1. Partial Least Squares Regression (PLSR)

The purpose of the PLSR analysis presented was to assess the effectiveness of the preprocessed VIS–NIR spectral data (410–990 nm) in non-destructively predicting both individual physicochemical attributes, namely moisture content (M.C.), water activity (a_w_), total soluble solids (TSSs), browning index (BI), color difference (ΔE), pH, and hardness (N), and the integrative quality index (Qi).

In the calibration process, the X-matrix comprised the second derivative of the absorbance spectrum, reflecting spectral profiles acquired from 15,000 date fruit samples measured monthly over the 12-month storage period. The corresponding Y-matrix included the reference values for the physicochemical variables and the Qi determined through standardized analytical protocols.

PLSR modeling was used to establish the correlations between the spectral features and each target variable. The predictive performance of the models was assessed through both calibration and cross-validation, using well-established metrics such as the coefficient of determination (R^2^), the root mean square error of calibration (RMSEC), and the root mean square error of cross-validation (RMSECV). Table 6 presents PLSR’s performance in the calibration analysis and cross-validation for the physicochemical properties of the stored date fruits (M.C., aw, TSS, BI, ΔE, pH, and hardness (N)), as well as the Qi.

In the calibration models, the R^2^ and RMSEC (mmd^−1^) values were 0.834, 0.817 and 0.833, 0.834 for M.C.%; 0.830, 0.813 and 0.819, 0.820 for a_w_; 0.832, 0.815 and 0.822, 0.823 for TSSs%; 0.821, 0.804 and 0.789, 0.790 for BI; 0.824, 0.807 and 0.799, 0.800 for ΔE; 0.814, 0.797 and 0.812, 0.813 for pH; 0.815, 0.798 and 0.853, 0.854 for hardness (N); and 0.790, 0.773 and 0.320, 0.321 for the Qi for the Sukkary dates and Khlass dates, respectively.

In terms of the cross-validation performance, the R^2^ and RMSECV values were 0.976, 0.959 and 0.756, 0.757 for M.C.%; 0.972, 0.955 and 0.742, 0.743 for aw; 0.974, 0.957 and 0.745, 0.746 for TSSs%; 0.963, 0.946 and 0.712, 0.713 for BI; 0.966, 0.949 and 0.722, 0.723 for ΔE; 0.956, 0.939 and 0.735, 0.736 for pH; 0.957, 0.940 and 0.776, 0.777 for hardness (N); and 0.932, 0.915 and 0.243, 0.244 for the Qi for the Sukkary dates and Khlass dates, respectively.

The cross-validation results yielded high coefficients of determination (R^2^ values), ranging from 0.932 to 0.976 for the Sukkary dates and 0.915 to 0.959 for the Khlass dates, reflecting strong predictive accuracy and the model’s robustness. These values far exceed the generally accepted threshold of 0.70 for a satisfactory NIR modeling performance, as noted in the prior literature [139,140]. Comparable studies applying partial least squares regression (PLSR) to fruits, such as cherries, apricots, apples, and kiwifruit, have reported similar or slightly lower R^2^ values, supporting the effectiveness of PLSR for evaluating internal quality traits through non-destructive means [141,142,143]. For instance, Carlini et al. (2000) achieved R^2^ values of up to 0.94 in predicting the soluble solid content in stone fruits by using VIS-NIR [141], while Peirs highlighted the influence of biological variability on the model performance for apples [142]. Similarly, Schaare and Fraser compared spectroscopic modes for assessing kiwifruit’s internal attributes, obtaining robust R^2^ values of around 0.90 [143]. Compared to these findings, the current models for Sukkary and Khlass dates exhibit a superior or comparable accuracy, underscoring the reliability of PLSR as a predictive tool for both physicochemical properties and sensory evaluations in stored date fruits.

#### 3.3.2. Artificial Neural Network (ANN) Analysis

The artificial neural network (ANN) model was trained using preprocessed second-derivative VIS–NIR spectral data (spanning 410–990 nm) as the input features, while the corresponding experimentally derived quality index (Qi) values served as the target outputs for supervised nonlinear regression. To evaluate the generalization performance, the trained model was validated on an independent test set, and its predictive accuracy was assessed using key performance metrics, including the coefficient of determination (R^2^) and the root mean square error of prediction (RMSEP).

Table 7 presents the performance of the ANNs in the calibration analysis and cross-validation for the physicochemical properties of the stored date fruits, including moisture content (M.C., %), water activity (a_w_), total soluble solids (TSSs, %), browning index (BI), color difference (ΔE), pH, and hardness (N), as well as the Qi characteristics.

To further support the quantitative evaluation, Figure 10 compares the observed versus the ANN-predicted Qi values for both the Sukkary and Khlass cultivars. This visual comparison enhances interpretability by demonstrating the close alignment between the predicted and actual values, thereby reinforcing the robustness of the ANN approach in modeling fruit quality.

In the calibration models, the R^2^ and RMSEC (mmd^−1^) values were 0.988, 0.971 and 0.846, 0.847 for M.C.%; 0.984, 0.967 and 0.832, 0.833 for aw; 0.986, 0.969 and 0.835, 0.836 for TSSs%; 0.975, 0.958 and 0.802, 0.803 for BI; 0.978, 0.961 and 0.812, 0.813 for ΔE; 0.968, 0.951 and 0.825, 0.826 for pH; 0.969, 0.952 and 0.866, 0.867 for hardness (N); and 0.944, 0.927 and 0.333, 0.334 for the Qi for the Sukkary and Khlass dates, respectively.

In terms of the cross-validation performance, the R^2^ and RMSECV values were 0.896, 0.969 and 0.816, 0.817 for M.C.%; 0.982, 0.965 and 0.802, 0.803 for aw; 0.984, 0.967 and 0.805, 0.806 for TSSs%; 0.973, 0.956 and 0.772, 0.773 for BI; 0.976, 0.959 and 0.782, 0.783 for ΔE; 0.966, 0.949 and 0.795, 0.796 for pH; 0.967, 0.950 and 0.836, 0.837 for hardness (N); and 0.942, 0.925 and 0.303, 0.304 for the Qi for the Sukkary and Khlass dates, respectively.

These findings confirm that the artificial neural network (ANN) model is highly effective in predicting the properties of SDFs. In particular, the ANN model outperformed the PLSR model, as demonstrated by its superior correlation coefficients, especially in relation to the normalized Qi parameter. These results align with earlier research that highlighted ANNs’ ability to model complex, nonlinear relationships with high predictive accuracy for foods [133,144,145,146,147,148,149,150,151]. For instance, Al-Rawahi et al. employed an ANN to evaluate the moisture sorption and phenolic content in dried pomegranate peels, achieving precise prediction outcomes [144]. Similarly, Goyal and Huang et al. emphasized the advantages of ANNs in modeling the physicochemical and sensory characteristics in food systems, citing improved generalization and accuracy compared to those with the traditional regression models [148]. Moreover, Przybył et al. successfully applied an ANN to quality-based classification of spray-dried juice powders, confirming the model’s robustness for predicting product quality attributes [151]. Collectively, these studies support the present results, reinforcing ANNs as a superior modeling approach to predicting both objective and subjective quality parameters in date-based functional food ingredients like SDFs.

Therefore, under the conditions of this study, it is recommended to use the ANN technique for the Qi (R^2^ = 0.942, 0.925) rather than the PLSR model (R^2^ = 0.932, 0.915) for Sukkary dates and Khlass dates, respectively.

One trend observed in the modeling process was that the root mean square error of calibration (RMSEC) consistently decreased with the addition of more components, indicating an improved fit to the calibration data. However, the root mean square error of cross-validation (RMSECV), which is a more reliable indicator of the model’s predictive power on unseen data, tended to increase beyond an optimal number of components. This highlights a common challenge in multivariate modeling: overfitting, where excessive model complexity can reduce the generalizability. Both the PLSR and ANN models demonstrated significant value in handling VIS-NIR spectral data, but the RMSECV remains the preferred metric for assessing real-world model performance. These findings align with previous research by Elamshity and Alhamdan, who used VIS-NIR spectroscopy and ANNs to evaluate date-syrup-based milk drinks and emphasized the RMSECV as a critical validation criterion in ANN-based calibration models [30]. Similarly, the RMSEC has been reported to reflect how well a model fits the training data, as the RMSECV provides a more accurate projection of the future predictive performance, especially when working with complex food matrices [152]. Therefore, despite the trade-off between fitting and generalizability, both modeling approaches remain effective when carefully optimized for spectral feature selection and component tuning.

Thus, both PLSR and ANN techniques have proven to be powerful tools for modeling NIR data in terms of calibration and cross-validation. Moreover, the ANN model is preferable to PLSR due to its higher correlation coefficients. Therefore, these analyses show that NIR results correlate well with the Qi and consequently with the individual properties of the SDFs.

In principle, the Qi can be determined during storage and in retail up to the shelf life of the SDFs set by the standard authorities [58]. For practical applications, food producers, processors, and authorities can effectively use a non-destructive, handheld NIR spectrophotometer throughout the entire production, processing, transportation, storage, and retail chain to monitor the “quality” and “shelf life” of produce.

#### 3.3.3. An Evaluation of the Prediction Error Metrics for the Chemometric Models

In addition to the conventional model assessment metrics (R^2^, RMSEC, RMSECV), three additional figures of merit were calculated to give a more complete evaluation of the predictive accuracy: the Root Mean Square Error of Prediction (RMSEP), the Relative Error of Prediction (REP), and the Range Error Ratio (RER). The RMSEP indicates the absolute size of the prediction errors, while the REP expresses this error as a percentage of the sample mean, providing a normalized view. The RER, on the other hand, measures the model’s ability to distinguish variations across the full range of reference values. Collectively, these metrics provide a deeper understanding of each model’s effectiveness in non-destructively predicting the quality index (Qi) across different date cultivars and storage conditions. Their use to assess model robustness and applicability to independent datasets are widely supported in chemometric literature.

Table 8 presents the prediction error metrics for the PLSR and ANN models. For the PLSR model, the RMSEP values were 0.256 for Sukkary dates and 0.266 for Khlass dates, corresponding to REP values of 8.12% and 8.75%, respectively. Although these values fall within the acceptable performance range for NIR-based models, the RER values (~10) suggest only moderate predictive strength [153]. In contrast, the ANN model achieved slightly higher RMSEP values (0.304 for Sukkary dates and 0.317 for Khlass dates) and yet exhibited a superior relative performance, with lower REP values (6.22% and 6.85%) and notably higher RER scores (13.2 and 12.7), indicating excellent prediction quality [133,154,155,156,157]. These metrics align with the accepted standards, where RER > 10 and REP < 10% denote highly reliable predictive models for food quality and spectral analysis applications [117,158].

#### 3.3.4. The Comparative Evaluation of the Predictive Accuracy Metrics

To contextualize the performance of our proposed models, we compared the REP, RMSEP, and RER values with those reported in prior VIS–NIR- or NIR-based studies applied to various horticultural products (Table 9. Comparative evaluation of prediction metrics).

The ANN model for the Sukkary dates achieved a REP of 6.22% and an RER of 13.2, which are well above the generally accepted thresholds for high predictive reliability in food quality modeling [23,148,156]. Comparable studies in green tea, apples, and apricots have typically reported REP values between 7% and 11%, with the RER values ranging from 8 to 12, depending on the spectral preprocessing and target analyses [23,141,156,159].

Although the PLSR model yielded slightly lower RMSEP values (0.266 for Khlass dates), the corresponding RER was below 10, suggesting a moderate predictive performance, consistent with reports highlighting the limitation of linear models in capturing the complex biochemical variability in perishable fruits [141,159].

These findings reinforce that the ANN model, trained on second-derivative VIS–NIR spectra within the 410–990 nm range, demonstrates a superior capacity to model the nonlinear relationships between spectral signatures and quality indicators. This observation is consistent with previous studies indicating that artificial neural networks tend to outperform traditional linear models, particularly when applied to complex, multivariate biological systems [23,33,148].

### 3.4. Evaluation of the Physicochemical Properties of the Date Fruits During Storage

The physicochemical properties (including mass (m), moisture content (M.C.), water activity (aw), browning index (BI), total color difference (∆E), potential of hydrogen (pH), and hardness) and sensory and microbial properties were measured and assessed for the Sukkary and Khlass dates. The effect of the storage duration, temperature, and packaging on each of these properties will be presented. Detailed data for the variance analysis, depicting the influence of these parameters on the objective and subjective characteristics of the stored Sukkary and Khlass date fruits, are available upon request from the corresponding author.

Figure 11 illustrates these data by showing the impact of storage time and packaging type on the properties of the dates (group A) stored at –18 °C. The figures displayed depict the changes in the key physicochemical properties, including mass, density, moisture content, water activity, browning index (BI), total color difference (ΔE), pH, and hardness, over a 12-month storage period, highlighting the influence of packaging type and storage duration.

Figure 11a demonstrates a consistent decrease in the mass of fruits for all packaging types during storage, with the steepest decrease observed in the OCCs and CCCs, particularly for the Khlass dates. These packaging materials offer minimal protection from moisture loss, resulting in partial drying. In contrast, the SSPCs and PSSPCs show the lowest decrease in mass due to their better sealing properties, which reduce water migration and minimize the exposure of the dates to the external environment.

The current results align with previous studies which emphasized the significance of the appropriate packaging in maintaining the physical integrity and quality of dates during storage. Effective packaging helps minimize moisture loss, a key factor influencing the texture, weight, and overall consumer acceptability of dates. Aleid et al. reported that optimized postharvest practices, including moisture barrier packaging, are essential for prolonging the shelf life and maintaining the marketability of dates under varying environmental conditions [160]. Similarly, Al-Abdoulhadi highlighted that inadequate packaging can accelerate dehydration and shrinkage, negatively impacting the visual and sensory attributes of stored dates [161]. These findings reinforce the present observations and underscore the critical role of packaging materials and design in preserving the physical and commercial quality of dates over extended storage periods.

The present study further confirms the effectiveness of specific packaging types, CCCs and CCSPBs, in minimizing mass loss during storage. These findings are consistent with previous research [160] which demonstrated that the use of modified-atmosphere packaging (MAP) and sealed packaging systems significantly reduces moisture migration and weight loss, thereby enhancing postharvest stability in dates. Additionally, the positive outcomes observed under cold storage conditions at 5 °C and −18 °C support earlier findings [161] which highlighted the role of low temperatures in maintaining physicochemical attributes, particularly mass retention and textural integrity, during extended storage periods. Together, these results reinforce the importance of selecting suitable packaging systems and storage temperatures to preserve the quality and commercial value of date fruits throughout the supply chain.

Figure 11b reveals that the density of the dates decreases gradually over time, with similar trends across all packaging types. The decrease in density is more pronounced in the OCCs and CCCs, as these materials allow for greater water loss and structural shrinkage. Comparatively, SSPCs and PSSPCs exhibit better density retention. The research by [160] supports the observation that plastic-based packaging, like shrink wrap, can preserve moisture better, reducing changes in density. In contrast, the OCC packaging, which allows for more air and moisture exchange, has higher density values due to it drying out. The Sukkary dates tended to show a slightly higher density than the Khlass dates, possibly related to their inherent texture and moisture content. The fresh Sukkary dates have an initial density (1.210 g/cm^3^ to 1.127 g/cm^3^) that is much higher than that observed after storage, suggesting that the fruit’s density naturally decreases due to water loss or the crystallization of sugars over time. Previous work [162] found that cultivars like the Sukkary cultivar have denser textures due to their fibrous content, which supports the slight density differences between the Khlass and Sukkary dates observed here.

As shown in Figure 11c, the moisture content significantly reduces overtime for all packaging types. CCCs show the sharpest decrease, particularly for the Khlass dates, indicating higher desiccation rates. In contrast, the SSPCs and PSSPCs retain the moisture content more effectively by providing a better barrier to external environmental factors. Similarly, Figure 11d demonstrates a gradual decrease in water activity over time, with PSSPCs and SSPCs outperforming the other packaging types in minimizing this loss. Lower water activity is crucial for reducing microbial activity and extending shelf life. In contrast, OCCs and CCCs, with higher rates of water activity, are more prone to accelerated microbial spoilage and textural degradation. These findings align with the literature emphasizing the importance of sealed and airtight packaging in preserving moisture, reducing water activity, and maintaining dates’ sensory and physicochemical properties. Additionally, studies have highlighted that combining airtight packaging with freezing storage conditions (−18 °C) is most effective in minimizing microbial risks and preserving quality during prolonged storage. Research indicates that higher moisture levels correlate with an increased risk of spoilage and microbial growth, which aligns with findings that a higher water activity (≥0.6) facilitates such risks [163,164]. PSSPCs consistently showed higher fruit moisture levels across most of the temperature groups. The Sukkary cultivar displayed a lower moisture content than that in the Khlass cultivar across all conditions, indicating the potential inherent stability or lower hygroscopicity of Sukkary dates. Previous studies date storage suggests similar trends regarding moisture content and water activity.

The browning index (BI), as shown in Figure 11e, increases over time for all packaging types, with CCSBPs exhibiting the highest BI for the Sukkary dates, indicating more intense browning, probably due to oxidative or enzymatic reactions. In contrast, the CCCs and SSPCs show relatively lower BI values, while PSSPCs demonstrate the least browning, reflecting better preservation of color. Similarly, Figure 11f illustrates a gradual increase in the total color difference (ΔE) across all packaging types, with CCSBPs showing the most significant changes, followed by SSPCs and PSSPCs. At the same time, the OCCs and CCCs exhibit less pronounced changes. These findings align with the literature, which emphasizes that sealed and airtight packaging effectively minimizes oxidation and browning, preserving both the color and visual appeal of dates, which is crucial for maintaining quality, consumer acceptance, and marketability [165].

The pH values shown in Figure 11g remain relatively stable across most of the packaging types, except the CCCs, in which they gradually decrease. The observed increase in acidity during prolonged storage is consistent with findings from previous studies, which attribute this trend to microbial activity and enzymatic transformations occurring over time. Similar patterns have been reported by Baltazari et al., who found that extended storage of citrus fruits led to elevated acidity levels, primarily due to the breakdown of sugars and increased microbial metabolism, effects that were more pronounced under suboptimal storage conditions [166]. Likewise, Dixit et al. observed comparable outcomes in tomatoes, where both the packaging material and storage duration influenced the accumulation of acids, resulting in noticeable changes in the fruit’s taste and quality [167]. These findings align with the present study’s results, emphasizing that acid buildup is a common physiological response in perishable fruits like dates during long-term storage, especially when the temperature and humidity are not tightly controlled. In contrast, PSSPCs demonstrate the highest pH stability throughout the storage period, supporting studies that have emphasized the role of airtight packaging in preserving the chemical stability of stored fruits.

The hardness values shown in Figure 11h illustrate that the Sukkary fruits are generally harder compared to the Khlass dates across all packaging types.

Additionally, Figure 12a,b highlights the impact of packaging type on the hard values throughout the storage period, revealing that the values for the PSSPCs remain relatively stable, demonstrating their ability to preserve the textural integrity of the dates. In contrast, in the other packaging types, such as OCCs and CCSBPs, they exhibit a minor decrease in hardness, which might be due to moisture loss or structural degradation. The effect of storage time on date quality, particularly hardness properties, has been extensively studied. Alfarsi et al. [168] reported a significant reduction in hardness over prolonged storage periods, attributing softening to the migration of moisture and enzymatic activity. Similarly, dates stored at ambient temperatures exhibited faster textural degradation compared to that in those stored at controlled low temperatures [169]. Additionally, the hardness of the dates decreased significantly after 6 months of storage, with variations depending on the cultivar and packaging type [170]. These findings underscore the importance of optimizing the storage conditions to maintain the sensory quality of dates.

Figure 13 illustrates the changes in the moisture content (%) of Sukkary and Khlass dates stored in pressed semi-rigid plastic containers (PSSPCs) under three storage temperatures (25 °C, 5 °C, and −18 °C) for 12 months. A significant decrease in the moisture content is observed for the dates stored at 25 °C, indicating greater loss of moisture compared to that at other temperatures. The higher temperature accelerates the diffusion of water vapor through the packaging material, leading to moisture loss. At 5 °C, the moisture content shows moderate stability with minimal changes in the PSSPC.

At −18 °C, the moisture content of the dates remained relatively stable throughout the 12-month storage period, indicating the effectiveness of low-temperature storage in preserving internal water levels. However, this stability was preceded by a noticeable initial decline in the moisture content during the early storage phase. This early reduction may be attributed to surface-level dehydration or the redistribution of unbound water as the fruit adjusts to subfreezing conditions. Once equilibrium was established, the further moisture loss was minimal, suggesting that the freezing environment successfully limited long-term degradation. This trend highlights the importance of accounting for both immediate and extended effects when evaluating cold storage performance.

Thus, the preservation efficacy at lower temperatures (5 °C and −18 °C) reduces the rate of moisture loss over time.

Sukkary fruits tend to lose moisture more rapidly at high temperatures, leading to a noticeable decline in their quality compared to that for the Khlass cultivar. The Khlass dates exhibit better resistance to moisture loss, even at 25 °C. Moisture stability at lower temperatures highlights the importance of refrigeration or freezing in maintaining the dates’ initial properties.

The current findings are in strong agreement with those from previous studies, indicating that elevated storage temperatures significantly accelerate moisture loss in dates, adversely affecting their texture, weight, and overall quality. For instance, Ahmed et al. (2021) demonstrated that Barhi dates stored at higher temperatures exhibited faster degradation in their texture and bioactive compound retention compared to these properties in dates stored in cooler conditions [31]. Similarly, Abdelzaher Radwan et al. (2023) reported that improper temperature and packaging combinations led to increased weight loss and reduced marketability in Saidy dates [170]. Comparable findings have also been observed in other tropical and sub-tropical fruits, such as papaya and peach, where warmer storage environments led to heightened transpiration and quality deterioration [171,172]. Furthermore, Mortazavi et al. (2007) emphasized that controlling the storage atmosphere and using vacuum or modified-atmosphere packaging can mitigate these losses in Khalal-stage dates [173], a conclusion also supported by Ramadan et al. (2020) in their study on Barhi dates stored in cold conditions [174]. Collectively, these studies reinforce the present conclusion that lower temperatures and the proper packaging are essential to slow moisture loss and preserve postharvest quality in date fruits.

Dates stored at freezing temperatures showed negligible changes in their moisture, preserving their texture and taste effectively. The literature also highlights the role of packaging materials like PSSPCs in reducing the moisture loss by providing a semi-barrier against diffusion, especially when combined with low storage temperatures [175]. Other works from the literature emphasize the importance of the storage conditions, noting that the optimal storage temperatures (4–10 °C) effectively preserve date quality over extended periods [176]. This highlights the need for the optimal temperature conditions and packaging to maintain moisture retention as a significant parameter in the quality and shelf life of stored dates.

### 3.5. Assessment of Microbial Quality

Across both cultivars and all storage conditions, the initial water activity/moisture content and the packaging barrier performance were the primary determinants of the microbial load. The lots with higher moisture (group E) consistently exhibited the highest TVC and YE, whereas the low-moisture controls showed markedly lower counts, confirming the protective effect of reduced water availability. Table 10 presents the total viable count (TVC) and yeast enumeration (YE) count in the Sukkary dates under different moisture contents and storage durations at various temperatures (−18 °C, +5 °C, and +25 °C) for the packaging type PSSPC. For the Sukkary dates in group E, the control lots rose in their ~TVCs from 1.1 to 4.5 log CFU g^−1^ at −18 °C, 2.0 to 7.0 at +5 °C, and 3.5 to 8.9 at +25 °C over 12 months, respectively, with the YE showing similar increases (0.9 → 3.9, 1.8 → 6.4, and 3.2 → 8.1 log CFU g^−1^, respectively). The Khlass dates displayed the same ordering at each time point. Overall, using the PSSPC at −18 °C was the most effective combination for minimizing microbial proliferation, while the OCCs/CCCs at +25 °C presented the greatest risk.

Table 11 displays the total viable count (TVC) and yeast enumeration (YE) count in the Khlass dates with different moisture contents at a constant temperature of +25 °C for the packaging type CCC over varying durations: as for the control group, both the TVC and YE counts increased with longer storage durations. For each storage duration, the higher-moisture-content groups (groups C to E) consistently exhibit higher TVCs and YE counts compared to those in the lower-moisture-content groups (groups A and B). These results are consistent with previous research studies, including those by [177,178,179].

Figure 14 shows the TVC and YE of the Sukkary and Khlass dates in different packaging types at +25 °C (group E) upon the completion of a 12-month storage period. At the 12-month endpoint, under the most permissive conditions (+25 °C, group E highest moisture), the microbial loads drop monotonically with an increasing package barrier. Open cartons (OCCs) show the largest counts, covered cartons (CCCs) are modestly lower, adding a sealable liner (CCSPB) produces a marked reduction, and rigid formats especially the pressed semi-rigid plastic container (PSSPC) yield the lowest loads. Numerically, the total viable counts (TVCs) for Sukkary dates decline from ~4.5 to ~0.7 log CFU g^−1^ when moving from the OCC to the PSSPC, with the yeast enumeration (YE) dropping from ~3.9 to ~0.2 log CFU g^−1^. Khlass dates follow the same rank order but at higher levels (TVC ~5.3 → ~1.6; YE ~4.5 → ~1.3 log CFU g^−1^), indicating cultivar-dependent susceptibility at ambient temperatures. These patterns are consistent with the expected mechanisms: the greater gas and moisture exchange in fiberboard (OCC/CCC) versus the restricted ingress and smaller headspace leakage paths in lined and rigid packs (CCSPBs/SSPCs/PSSPCs). Practically, the data show that high-barrier packaging is essential for high moisture dates stored at room temperature, though it does not fully eliminate growth underscoring the need to control the initial moisture and, where feasible, use colder storage. The ANOVA with the LSD test (*p* < 0.05) confirmed that PSSPCs (and typically CCSPB/SSPC) had significantly lower TVCs and YE than those in the OCCs/CCCs at this endpoint, supporting the recommendation to avoid low-barrier cartons for ambient distribution of moist date fruits, in line with prior work on barrier-driven microbial stability in dates [177,178,179].

Figure 15 shows the TVC and YE in the Sukkary dates at different storage temperatures (group E) for packaging type PSSPC upon the completion of a 12-month storage period. Using the highest-barrier packaging (PSSPC) and the highest-moisture lots (group E), Figure 15 isolates the effect of temperature after 12 months. Both the total viable counts (TVCs) and yeast enumeration (YE) increase monotonically as follows fresh (as-harvested control (zero time)) → −18 °C → +5 °C → +25 °C confirming that temperature control dominates the microbial outcomes even under high-barrier packaging. Freezing (−18 °C) keeps the TVC/YE close to the baseline (~1–2 log CFU g^−1^); refrigeration (+5 °C) slows but does not prevent growth (~4 log CFU g^−1^); and room temperature (+25 °C) yields the highest loads (≈6–7 log CFU g^−1^). Khlass dates show consistently higher values than Sukkary dates at each temperature (≈0.3–0.9 log), indicating cultivar-dependent susceptibility under high moisture. The yeast trends mirror those for the TVC, implying a broad microbial response to temperature rather than a single group effect. The ANOVA with the LSD test (*p* < 0.05) confirmed the significant differences across temperatures for both cultivars. Practically, these data show that while the PSSPC mitigates ingress and headspace effects, the barrier performance cannot substitute for a cold chain; for moist dates, −18 °C is required to maintain low counts over prolonged storage, whereas +25 °C rapidly erodes the microbial stability, findings consistent with prior work on the temperature-driven spoilage kinetics in dates [180,181,182].

The current findings reinforce the well-documented importance of monitoring the microbial counts in dates during storage, as microbial proliferation can significantly affect their safety, shelf life, and marketability. Several prior studies have emphasized that factors such as moisture content, packaging type, and storage temperature are key determinants of microbial growth in stored date fruits [183,184,185,186]. For example, Al-Yahyai and Khan highlighted the necessity of implementing hygienic postharvest practices in Oman’s date industry to prevent microbial contamination during storage and handling [183]. In a foundational text, Doyle and Beuchat (2007) underscored the susceptibility of high-sugar, low-acidity fruits like dates to spoilage microorganisms, particularly when stored under poor sanitary or temperature conditions [184]. Moreover, Hamad documented significant microbial spoilage in Rutab-stage dates sold in Saudi markets, noting elevated counts of yeasts and molds that compromised fruit quality and safety [186]. Taking together, these studies confirm the present observation that controlling the microbial load is critical to maintaining postharvest date quality and extending shelf life under varying storage scenarios. Details on the effect of microbial dynamics on the properties and physicochemical characteristics of diverse date varieties stored under the influence of moisture, packaging, and temperature fluctuations can be found in another study (Elamshity, 2025, under publication) [185].

### 3.6. Sensory Evaluation of Stored Date Fruits (SDFs) During Storage

The sensory evaluations of the stored date fruits (SDFs) were conducted by sixty-six participants, using a nine-point structured hedonic scale for texture, taste, hardness, color, adhesion, peelability, and overall acceptability. The sensory evaluation involved assessing samples of the SDFs under various temperature, duration, and packaging type conditions, resulting in the collection of 23.760 evaluation forms.

Standardizing the pre-tasting preparation (dry-cleaning, uniform pitting/cutting, and fixed portion sizes and temperature), blinding, and the randomized/counterbalanced serving order reduced the panel bias and within-treatment variability, improving the interpretability of the treatment effects across the packaging and temperature conditions. The one-way flow (no sample was returned to storage) and the safety gate minimized the risk of cross-contamination and ensured that only microbiologically acceptable samples were evaluated.

The findings showed that the fresh date fruits from group A, with a moisture content of 10.531% for Sukkary dates and 14.877% for Khlass dates at the beginning of the experiments, were favored over the other moisture content groups. At the end of storage, the SDFs from pressed semi-rigid plastic containers (PSSPCs) stored at −18 °C were significantly (α = 0.05) preferred for all moisture content groups. Detailed data on the sensory evaluations of the stored Sukkary and Khlass dates are available on request from the corresponding author.

Figure 16 summarizes the results of the sensory evaluation, with the standard deviation bars representing the inter-panelist variability. For both cultivars, samples stored at −18 °C in pressed semi-rigid plastic containers (PSSPCs) achieved the highest preference scores across multiple sensory attributes, including taste, texture, peelability, and overall acceptability. The limited overlap in the error bars between these samples and the lower-scoring groups suggests that the differences are likely statistically meaningful and not merely due to random variation. Conversely, attributes such as color and adhesion exhibited greater overlap in the error margins, indicating that the differences observed in these traits may fall within normal sensory variability and may not reach statistical significance. These findings collectively underscore the role of low-temperature storage and high-barrier packaging in maintaining sensory quality during long-term storage, as emphasized in the sensory science literature [77,79,187,188,189].

For clarity, the Sukkary set comprised a moisture content control (Control-SUK) and fresh samples from groups A–E (SUK-A-Fresh… SUK-E-Fresh). The same groups were then stored in pressed semi-rigid plastic containers (PSSPC) at 5 °C, 25 °C, and −18 °C, with labels reflecting group and temperature (SUK-A-5-PSSPC, SUK-A-25-PSSPC, SUK-A-F18-PSSPC). The Khlass set followed the identical scheme with a control (Control-KHL) and fresh A–E groups (KHL-A-Fresh… KHL-E-Fresh), plus PSSPC-packed storage at 5 °C, 25 °C, and −18 °C using the parallel naming convention (KHL-A-5-PSSPC, KHL-A-25-PSSPC, KHL-A-F18-PSSPC). This standardized nomenclature is used throughout the discussion to map results unambiguously to cultivar, moisture group, temperature, and packaging.

### 3.7. The Application of the NIR and Qi Models to Subjective and Objective Assessments

Figure 17 presents the tendency of the quality index from the sensorial evaluation data (Qis) and the data from the objective browning index (OBI) tests for the stored date fruits to decrease. Both tests exhibit similar trends, showing a decrease in the subjective and objective data with storage. The correlation coefficients (R^2^) for the quality index (Qi) in cross-validation ranged from 0.932 to 0.986 for the Sukkary dates and 0.915 to 0.969 for the Khlass dates, demonstrating the excellent performance of the model.

The results of this study suggest a strong and reliable correlation between the sensory evaluations and objective measurements, particularly through the integration of the Qi index and the VIS-NIR spectral data. This correlation highlights the potential to apply such models in industrial automation systems aimed at quality assurance in date processing. These findings agree with previous research that supports the application of near-infrared spectroscopy (NIR) as a non-destructive, real-time tool for assessing food quality. Cen and He emphasized the theoretical foundation and practical advantages of NIR in capturing chemical and physical attributes linked to sensory characteristics in food products [153]. Likewise, Huang et al. reviewed the role of in-line and online NIR monitoring systems in the food and beverage industries, demonstrating their efficacy in correlating instrument readings with consumer-perceived quality [156]. In the context of fresh produce, Nicolaï et al. confirmed that NIR-based models could accurately reflect internal quality parameters such as sweetness, firmness, and acidity—key indicators often evaluated in sensory testing [159]. Together, these studies support the current conclusion that integrating spectral analyses with sensory-derived quality indices like Qi provides a promising framework for implementing automated quality control systems in date fruit processing lines.

### 3.8. Limitations and Considerations in Non-Destructive Spectroscopic Assessments of Date Fruits’ Shelf Lives

Notwithstanding the robust predictive capabilities of the proposed Qi framework, it is imperative to acknowledge several of its limitations. The model was calibrated using two cultivars (Sukkary and Khlass) under controlled storage conditions, which may restrict its generalizability to other varieties or dynamic supply chain environments [190]. Furthermore, although ANNs have been shown to outperform linear models, they are susceptible to overfitting and necessitate regular retraining to accommodate seasonal and compositional variations [191]. Furthermore, the spectral distortion caused by specific packaging types has been demonstrated to have the capacity to affect the accuracy of the predictions [156]. Moreover, the model is not currently equipped with economic variables, thus restricting its comprehensive application in commercial contexts [192]. Future research should concentrate on expanding the cultivar datasets, integrating multimodal data, and validating this approach under real-world conditions [193].

## 4. Conclusions

The quality index (Qi) developed in this study offers a fast, non-destructive, and reliable tool for assessing the quality of stored date fruits (SDFs) across storage and supply chains. VIS–NIR spectroscopy effectively predicted the Qi by analyzing the spectral responses and correlating them with key physicochemical traits such as moisture, aw, TSSs, pH, ΔE, hardness, and BI. Both the PLSR and ANN models showed strong predictive power, with the ANN model achieving R^2^ values of 0.932–0.986 (Sukkary) and 0.915–0.969 (Khlass). The highest Qi values were observed at –18 °C for the Sukkary dates in SSPC group C (MC: 16.60%) and the Khlass dates in CCC group D (MC: 32.06%). PSSPC packaging combined with freezing minimized microbial growth over 12 months. These results underscore the role of packaging, temperature, and moisture in preserving quality. This approach enables real-time monitoring of quality and self-life. VIS–NIR is suitable for implementation by regulators (SFDA.FD 150-1/2018) [58] and manufacturers and throughout production and distribution chains.

## Figures and Tables

**Figure 1 foods-14-03060-f001:**
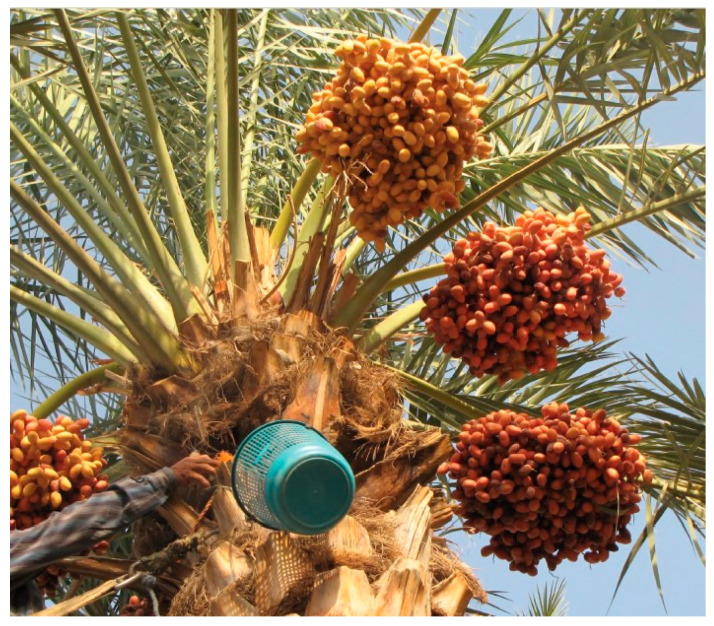
One of the date palm trees from which fruits were harvested for this study, located in the Derab region of Riyadh, Saudi Arabia (photograph taken by Mahmoud G. Elamshity, 2023).

**Figure 2 foods-14-03060-f002:**
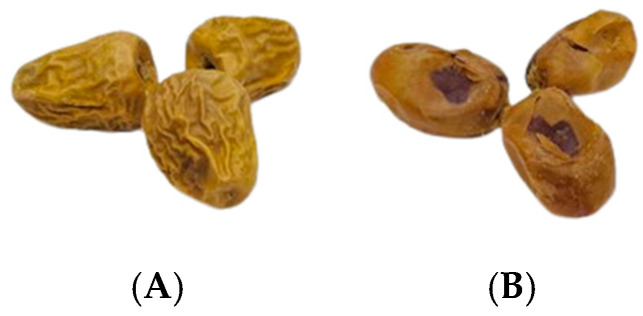
The date samples of (**A**) Sukkary and (**B**) Khlass fruits at the Tamr stage.

**Figure 3 foods-14-03060-f003:**
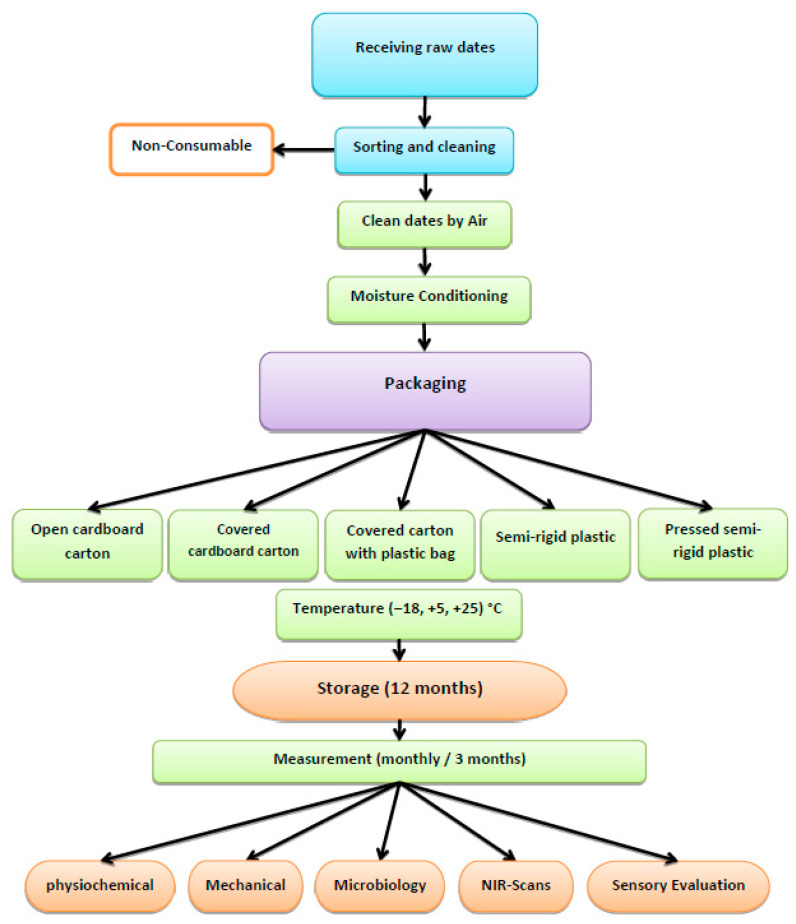
Experimental plans for the dates and measurements of their physicochemical, microbial, texture, and sensory assessments.

**Figure 4 foods-14-03060-f004:**
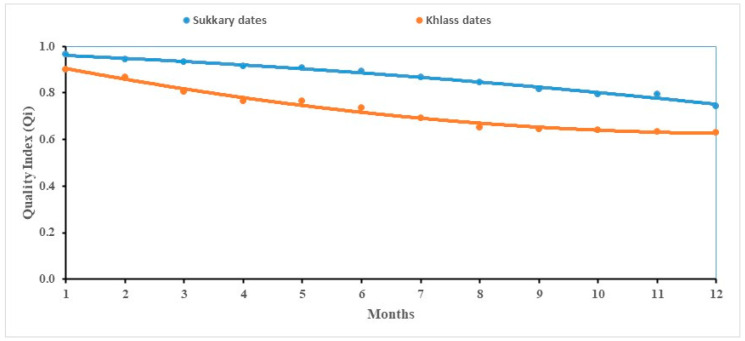
The time course of the normalized quality index (Qi) for stored date fruits both experimentally measured and predicted using VIS–NIR-based regression models for the Sukkary and Khlass cultivars.

**Figure 5 foods-14-03060-f005:**
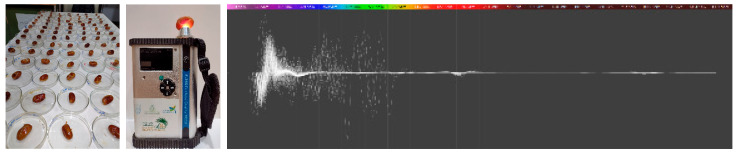
An image of the raw spectral absorbance of 15,000 samples of Sukkary and Khlass fruits under different storage conditions (**right**) utilizing the Flex F750 instrument (**middle**) and fruit samples (**left**).

**Figure 6 foods-14-03060-f006:**
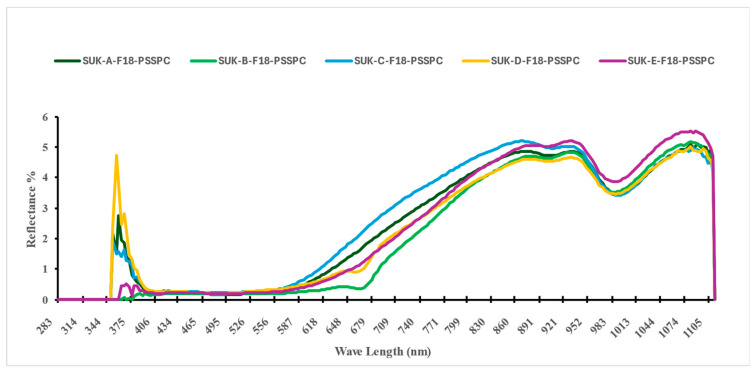
The reflectance spectra of the stored Sukkary date fruits at the end of the 12th month of storage.

**Figure 7 foods-14-03060-f007:**
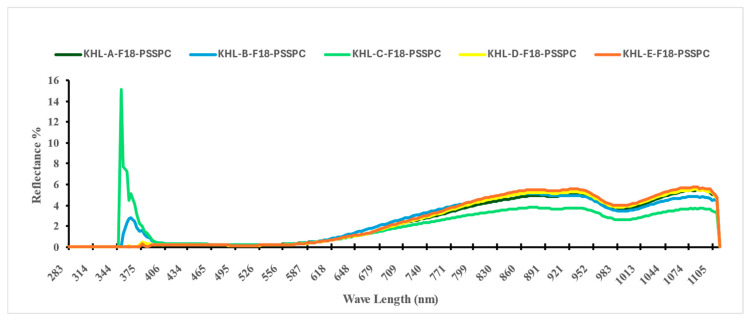
The reflectance spectra of the stored Khlass date fruits at the end of the 12th month of storage.

**Figure 8 foods-14-03060-f008:**
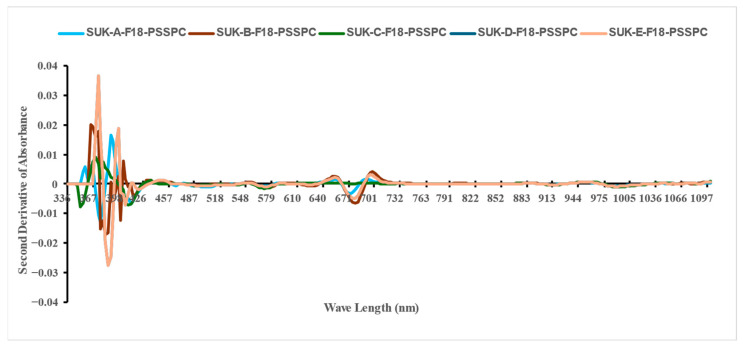
The average second derivative of the absorbance within the wavelength range of 327–1101 nm for the stored Sukkary date fruits at the end of the 12th month of storage.

**Figure 9 foods-14-03060-f009:**
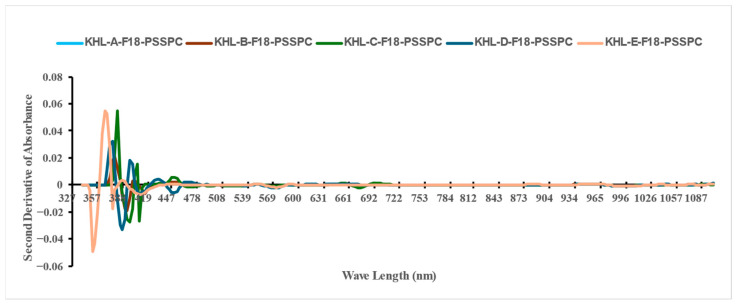
The average second derivative of the absorbance within the wavelength range of 327–1101 nm for the stored Khlass dates at the end of the 12th month of storage.

**Figure 10 foods-14-03060-f010:**
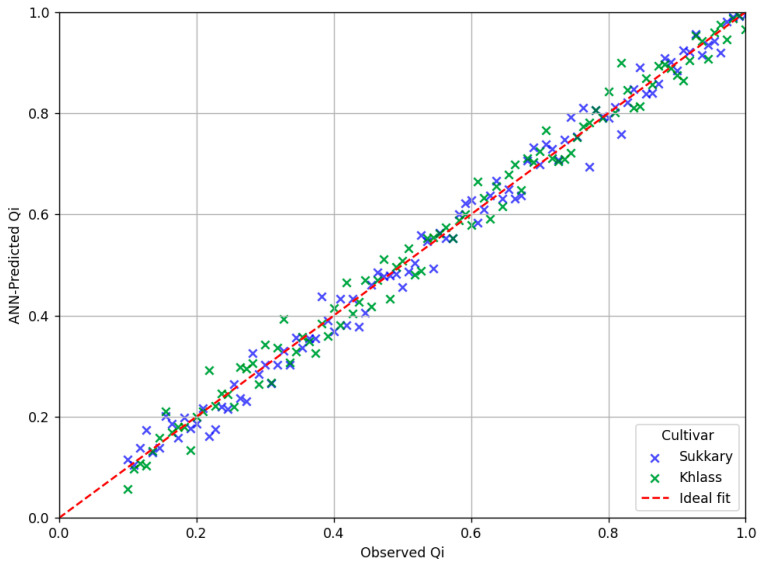
Comparison of observed and ANN-predicted quality index (Qi) for Sukkary and Khlass dates based on VIS–NIR spectral data.

**Figure 11 foods-14-03060-f011:**
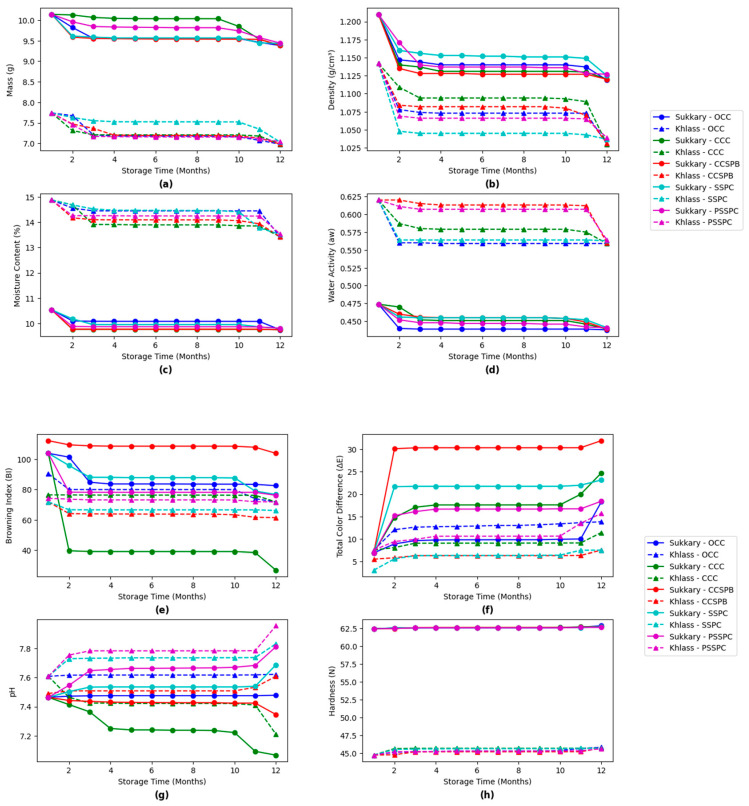
The effect of storage time and packaging on the properties of the date fruits (group A) stored at –18 °C. Figure (**a**) through (**h**) illustrate the changes in various quality parameters of dates over the storage period. Specifically, Figure (**a**) shows the relationship between mass and storage time (months), Figure (**b**) presents changes in density, Figure (**c**) displays moisture content, and Figure (**d**) shows water activity. Figure (**e**) depicts the browning index, Figure (**f**) represents the total color difference, Figure (**g**) illustrates pH variation, and Figure (**h**) shows changes in hardness as a function of storage duration.

**Figure 12 foods-14-03060-f012:**
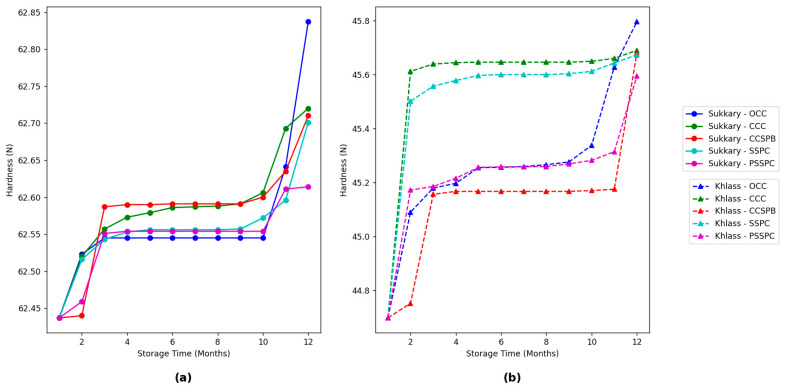
The effect of storage time and packaging on the hardness of (**a**) Sukkary and (**b**) Khlass dates (group A) stored at –18 °C.

**Figure 13 foods-14-03060-f013:**
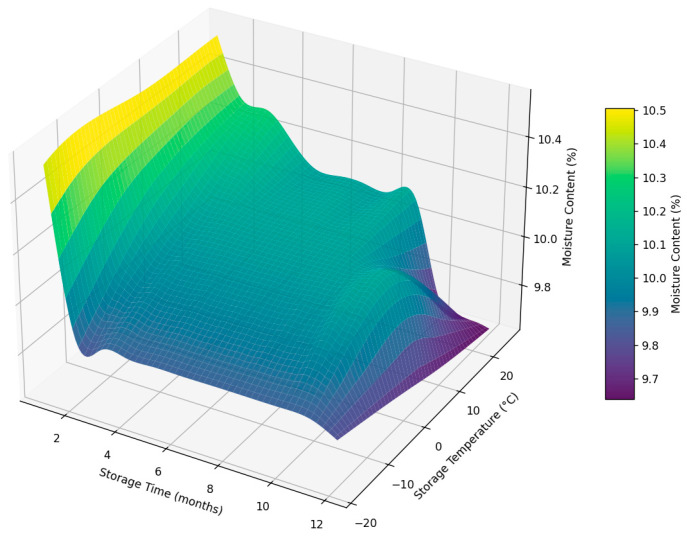
The effect of storage time and temperature on the moisture content of Sukkary and Khlass dates in pressed semi-rigid plastic containers (PSSPCs).

**Figure 14 foods-14-03060-f014:**
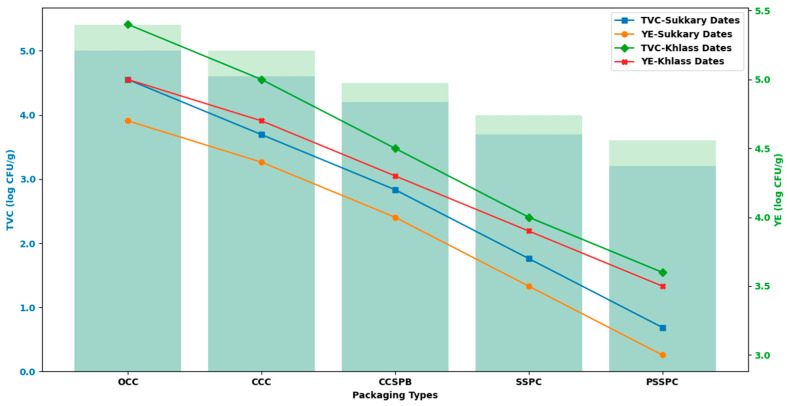
The TVC and YE in the Sukkary and Khlass dates with different packaging types at +25 °C (group E) upon the completion of the 12-month storage period.

**Figure 15 foods-14-03060-f015:**
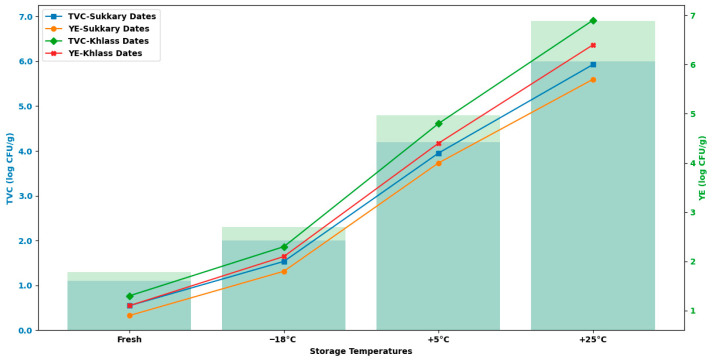
TVC and YE in Sukkary and Khlass dates at different storage temperatures (group E) for the packaging type PSSPC upon the completion of the 12-month storage period.

**Figure 16 foods-14-03060-f016:**
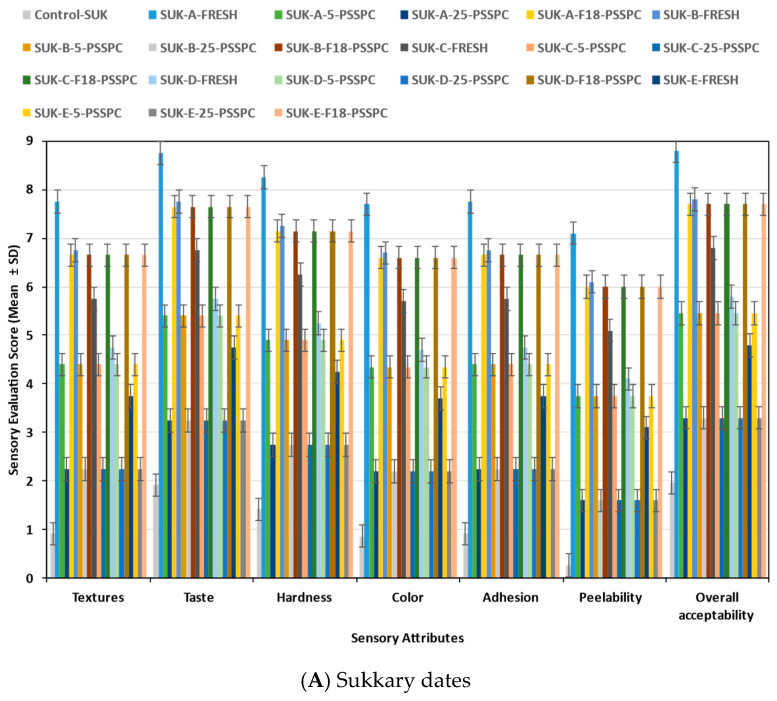
Sensory preference in texture, taste, hardness, color, adhesion, peelability, and overall acceptability for the stored date fruits in different packages at −18, 5, and 25 °C for (**A**) Sukkary dates, and (**B**) Khlass dates.

**Figure 17 foods-14-03060-f017:**
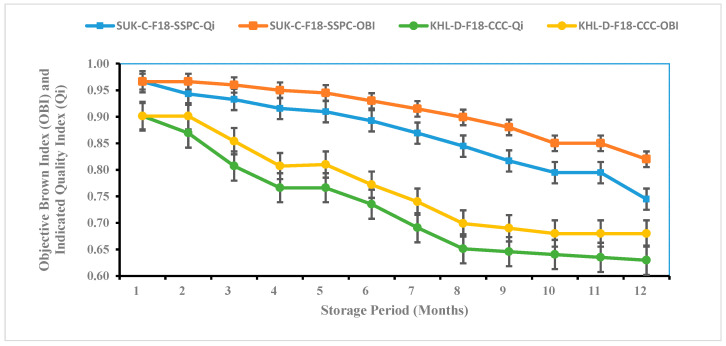
The quality index from the sensorial evaluation data (Qis) and that from the objective browning index (OBI) tests (*n* = 500 fruits for each storage period) for the stored date fruits (SDFs) (Sukkary, Khlass), ranging from freshness (0 days) to 12 months of storage.

**Table 1 foods-14-03060-t001:** Moisture content (d.b.%) classification of the postharvest Sukkary and Khlass dates across the grouped hydration levels *.

Class	Moisture Content (db.) (%)	
	Sukkary	Khlass
Control	4.357 ^f^ ± 0.015	6.235 ^f^ ± 0.021
Group A	10.531 ^e^ ± 0.035	14.877 ^e^ ± 0.023
Group B	13.181 ^d^ ± 0.011	21.781 ^d^ ± 0.017
Group C	16.606 ^c^ ± 0.014	26.807 ^c^ ± 0.036
Group D	20.489 ^b^ ± 0.054	32.062 ^b^ ± 0.046
Group E	27.343 ^a^ ± 0.016	38.697 ^a^ ± 0.025

* The same letter in a column within a group indicates that the average values are not significantly different according to the least squares difference (LSD) test at a significance level of *p* < 0.05.

**Table 2 foods-14-03060-t002:** Barrier properties and compliance documentation for packaging systems.

Packaging Materials	Construction /Polymer	Nominal Thickness/ Grammage	WVTR (g·m^−2^·day^−1^) @38 °C/90% RH	OTR (cm^3^·m^−2^·day^−1^·bar^−1^) @23 °C/0% RH	Material Code	Lot/Batch	* CoC/DoC (ID)	Food-Contact Compliance [46]
OCC	Corrugated Kraft paperboard	[440–600 g·m^−2^]	300–1500	>10,000 (not a gas barrier)	[Carton-200/112/200 B-flute]	[LOT 23B-4719]	[CoC #A-2024-118]	FDA 21 CFR (paper and board guidance), EU 1935/2004 (general) [47]
CCC	Paperboard with lid/overwrapping	[440–600 g·m^−2^]	250–1200	>10,000	[Carton-200/125/200 E-flute]	[LOT 23E-5120]	[CoC #A-2024-119]	FDA 21 CFR; EU 1935/2004 [47]
CCSPB	Paperboard + heat-sealable LDPE liner (≈50–70 µm)	LDPE: [50–70 µm]	5–15	2000–8000	[LDPE-60 µm grade]	[LOT 24-05-18-07]	[DoC #B-10/2011-221]	FDA 21 CFR 177.1520; EU 1935/2004; EU 10/2011 (plastics) [48]
SSPC	PET or PP tub (wall ≈ 0.4–0.8 mm)	PET/PP: [0.5–0.8 mm]	1–5	2–20 (PET lower; PP higher)	[PET 700 µm grade]	[LOT KSA-240713-02]	[CoC #C-177.1630-045]	FDA 21 CFR 177.1630 (PET)/177.1520 (PP); EU 1935/2004; EU 10/2011 [48]
PSSPC	Multilayer PET/EVOH/PE (co-extruded)	Wall: [0.5–0.8 mm]; EVOH core [~3–10%]	0.1–0.8	0.1–1.0	[PET/EVOH/PE 650 µm spec]	[LOT EVOH-241009-17]	[DoC #D-10/2011-508]	FDA 21 CFR; EU 1935/2004; EU 10/2011 (incl. specific migration) [49]

* CoC/DoC (IDs): supplier Certificate of Conformance/Declaration of Conformance.

**Table 3 foods-14-03060-t003:** Sensory panel release criteria (dates; ready-to-eat, low-moisture).

Analyte	Method and Matrix	Acceptance Limit	Investigate/Hold	Reject (Do Not Serve)
Total Viable Count (TVC)	ISO 4833-1 (aerobic plate count, 30 °C); dates, ready-to-eat [73]	≤5.0 log CFU g^−1^ (≤10^5^)	5.0–6.0 log CFU g^−1^	>6.0 log CFU g^−1^
Yeast Enumeration (YE)	ISO 21527-2 (yeasts and molds for low-aw foods); dates, ready-to-eat [74]	≤4.0 log CFU g^−1^ (≤10^4^)	4.0–5.0 log CFU g^−1^	>5.0 log CFU g^−1^

**Table 4 foods-14-03060-t004:** Constant (a, b, c) and correlation coefficient (R^2^) values from the prediction equation of the quality index (Qi) for the stored Sukkary and Khlass date fruits during storage (t, months).

Cultivar	Qi = at^2^ − bt + c			R^2^
	A	b	c	
Sukkary	−0.0007	0.0102	0.9719	0.988
Khlass	0.0021	0.0521	0.9554	0.984

**Table 5 foods-14-03060-t005:** Model fit comparison between linear and quadratic equations based on ANOVA and AIC.

Cultivar	Model Type	Correlation Coefficient (R^2^)	Akaike Information Criterion (AIC)	Analysis of Variance (ANOVA) *p*-Value
Sukkary	Linear	0.976	120.3	-
Sukkary	Quadratic	0.988	110.5	<0.05
Khlass	Linear	0.927	135.2	-
Khlass	Quadratic	0.984	119.8	<0.05

**Table 6 foods-14-03060-t006:** PLSR’s performance for M.C., a_w_, TSS, BI, ΔE, pH, hardness (N), and Qi (*n* = 1000 fruits for each property) with both calibration and cross-validation models of stored Sukkary and Khlass date fruits, respectively.

Cultivar	Parameter	Calibration		Cross-Validation	
		R^2^	RMSEC	R^2^	RMSECV
Sukkary	M.C._S_	0.834	0.833	0.976	0.756
	aw_S_	0.830	0.819	0.972	0.742
	TSS_S_	0.832	0.822	0.974	0.745
	BI_S_	0.821	0.789	0.963	0.712
	ΔE_S_	0.824	0.799	0.966	0.722
	pH_S_	0.814	0.812	0.956	0.735
	Hardness_S_	0.815	0.853	0.957	0.776
	Qi_S_	0.790	0.320	0.932	0.243
Khlass	M.C._K_	0.817	0.834	0.959	0.757
	aw_K_	0.813	0.820	0.955	0.743
	TSS_K_	0.815	0.823	0.957	0.746
	BI_K_	0.804	0.790	0.946	0.713
	ΔE_K_	0.807	0.800	0.949	0.723
	pH_K_	0.797	0.813	0.939	0.736
	Hardness_K_	0.798	0.854	0.940	0.777
	Qi_K_	0.773	0.321	0.915	0.244

**Table 7 foods-14-03060-t007:** The ANNs’ performance for M.C., a_w_, TSS, BI, ΔE, pH, hardness (N), and Qi (*n* = 1000 fruits for each property) with both calibration and cross-validation models for stored Sukkary and Khlass date fruits, respectively.

Cultivar	Parameter	Calibration		Cross-Validation	
		R^2^	RMSEC	R^2^	RMSECV
Sukkary	M.C._S_	0.988	0.846	0.986	0.816
	aw_S_	0.984	0.832	0.982	0.802
	TSS_S_	0.986	0.835	0.984	0.805
	BI_S_	0.975	0.802	0.973	0.772
	ΔE_S_	0.978	0.812	0.976	0.782
	pH_S_	0.968	0.825	0.966	0.795
	Hardness_S_	0.969	0.866	0.967	0.836
	Qi_S_	0.944	0.333	0.942	0.303
Khlass	M.C._K_	0.971	0.847	0.969	0.817
	aw_K_	0.967	0.833	0.965	0.803
	TSS_K_	0.969	0.836	0.967	0.806
	BI_K_	0.958	0.803	0.956	0.773
	ΔE_K_	0.961	0.813	0.959	0.783
	pH_K_	0.951	0.826	0.949	0.796
	Hardness_K_	0.952	0.867	0.950	0.837
	Qi_K_	0.927	0.334	0.925	0.304

**Table 8 foods-14-03060-t008:** Prediction error metrics for PLSR and ANN models *.

Model	Cultivar	RMSEP	REP (%)	RER
PLSR	Sukkary	0.256	8.12	10.6
PLSR	Khlass	0.266	8.75	9.8
PLSR	Combined	0.261	8.44	10.2
ANN	Sukkary	0.304	6.22	13.2
ANN	Khlass	0.317	6.85	12.7
ANN	Combined	0.310	6.54	13.0

* REP = (RMSEP/mean reference value) × 100; RER = (range of reference values/RMSEP).

**Table 9 foods-14-03060-t009:** Comparative evaluation of prediction metrics.

Study/Reference	Product/Model	RMSEP	REP (%)	RER	Notes
Current study (Sukkary, ANNs)	Date fruits/ANN	0.304	6.22	13.2	High precision, strong RER ≥ 10, low REP < 7% indicates excellent predictability
Current study (Khlass, PLSR)	Date fruits/PLSR	0.266	8.75	9.8	Moderate performance; just below the ideal RER threshold, acceptable error levels
Huang et al. (2008) [156]	Firmness of apples/PLSR	1.24	~9–11	~8.5	Moderate performance, good for bulk grading
Carlini et al. (2000) [141]	Soluble solids in apricot/PLSR	0.68	~10	~9.5	An early benchmark in VIS-NIR modeling
Zhang et al. (2021) [23]	Moisture in green tea/NIR	0.235	~7	12.1	High performance using NIR + ANNs in controlled processing settings
Nicolaï et al. (2007) [159]	Internal quality of peaches	0.290	8–10	~10	Review: High variability in fruit matrices affects the general RER
Goyal (2013) [148]	ANN in various fruits	–	6–9	>10	General trend: the ANN outperforms linear models, especially for nonlinear degradation

**Table 10 foods-14-03060-t010:** TVCs and YE counts in Sukkary dates under different moisture contents and storage durations at various temperatures (−18 °C, +5° C, and +25 °C) for packaging type PSSPC *, **.

Moisture Content	Duration	TVC	TVC	TVC	YE	YE	YE
		(log CFU/g)	(log CFU/g)	(log CFU/g)	(log CFU/g)	(log CFU/g)	(log CFU/g)
(%)	(Months)	−18 °C	+5 °C	+25 °C	−18 °C	+5 °C	+25 °C
Control (4.357)	3	1.1 ^d^ ± 0.21	2.0 ^d^ ± 0.02	3.5 ^d^ ± 0.03	0.9 ^d^ ± 0.01	1.8 ^d^ ± 0.01	3.2 ^d^ ± 0.02
	6	1.2 ^c^ ± 0.11	2.2 ^c^ ± 0.02	3.8 ^c^ ± 0.03	1.0 ^c^ ± 0.01	2.0 ^c^ ± 0.02	3.4 ^c^ ± 0.03
	9	1.3 ^b^ ± 0.21	2.4 ^b^ ± 0.02	4.0 ^b^ ± 0.13	1.1 ^b^ ± 0.01	2.2 ^b^ ± 0.02	3.6 ^b^ ± 0.03
	12	1.4 ^a^ ± 0.11	2.6 ^a^ ± 0.02	4.3 ^a^ ± 0.13	1.2 ^a^ ± 0.01	2.4 ^a^ ± 0.02	3.8 ^a^ ± 0.03
Group A (10.531)	3	1.6 ^d^ ± 0.31	3.0 ^d^ ± 0.02	4.7 ^d^ ± 0.13	1.3 ^d^ ± 0.01	2.6 ^d^ ± 0.02	4.2 ^d^ ± 0.03
	6	1.8 ^c^ ± 0.21	3.3 ^c^ ± 0.23	5.0 ^c^ ± 0.04	1.5 ^c^ ± 0.01	2.9 ^c^ ± 0.03	4.5 ^c^ ± 0.04
	9	2.0 ^b^ ± 0.12	3.6 ^b^ ± 0.13	5.3 ^b^ ± 0.04	1.7 ^b^ ± 0.02	3.2 ^b^ ± 0.13	4.8 ^b^ ± 0.04
	12	2.2 ^a^ ± 0.42	4.0 ^a^ ± 0.13	5.6 ^a^ ± 0.04	1.9 ^a^ ± 0.02	3.5 ^a^ ± 0.13	5.1 ^a^ ± 0.04
Group B (13.181)	3	1.9 ^d^ ± 0.22	3.4 ^d^ ± 0.23	5.2 ^d^ ± 0.04	1.6 ^d^ ± 0.12	3.1 ^d^ ± 0.03	4.7 ^d^ ± 0.24
	6	2.1 ^c^ ± 0.42	3.7 ^c^ ± 0.13	5.5 ^c^ ± 0.04	1.8 ^c^ ± 0.02	3.4 ^c^ ± 0.03	5.0 ^c^ ± 0.14
	9	2.4 ^b^ ± 0.42	4.0 ^b^ ± 0.03	5.8 ^b^ ± 0.14	2.0 ^b^ ± 0.02	3.7 ^b^ ± 0.03	5.3 ^b^ ± 0.14
	12	2.6 ^a^ ± 0.12	4.4 ^a^ ± 0.13	6.2 ^a^ ± 0.14	2.2 ^a^ ± 0.02	4.0 ^a^ ± 0.13	5.6 ^a^ ± 0.14
Group C (16.606)	3	2.3 ^d^ ± 0.02	4.2 ^d^ ± 0.13	5.9 ^d^ ± 0.24	1.9 ^d^ ± 0.02	3.8 ^d^ ± 0.03	5.4 ^d^ ± 0.24
	6	2.5 ^c^ ± 0.02	4.5 ^c^ ± 0.13	6.2 ^c^ ± 0.44	2.1 ^c^ ± 0.02	4.1 ^c^ ± 0.13	5.7 ^c^ ± 0.24
	9	2.8 ^b^ ± 0.02	4.9 ^b^ ± 0.13	6.5 ^b^ ± 0.24	2.4 ^b^ ± 0.02	4.4 ^b^ ± 0.03	6.0 ^b^ ± 0.24
	12	3.0 ^a^ ± 0.12	5.3 ^a^ ± 0.13	6.9 ^a^ ± 0.44	2.6 ^a^ ± 0.02	4.8 ^a^ ± 0.03	6.4 ^a^ ± 0.34
Group D (20.489)	3	2.7 ^d^ ± 0.22	4.8 ^d^ ± 0.13	6.6 ^d^ ± 0.44	2.3 ^d^ ± 0.02	4.3 ^d^ ± 0.03	5.9 ^d^ ± 0.24
	6	3.0 ^c^ ± 0.12	5.1 ^c^ ± 0.23	7.0 ^c^ ± 0.44	2.5 ^c^ ± 0.02	4.6 ^c^ ± 0.03	6.2 ^c^ ± 0.24
	9	3.4 ^b^ ± 0.32	5.5 ^b^ ± 0.13	7.4 ^b^ ± 0.74	2.8 ^b^ ± 0.02	5.0 ^b^ ± 0.03	6.6 ^b^ ± 0.24
	12	3.7 ^a^ ± 0.12	6.0 ^a^ ± 0.14	7.9 ^a^ ± 0.25	3.1 ^a^ ± 0.02	5.4 ^a^ ± 0.03	7.1 ^a^ ± 0.14
Group E (27.343)	3	3.6 ^d^ ± 0.32	5.8 ^d^ ± 0.04	7.6 ^d^ ± 0.05	3.0 ^d^ ± 0.12	5.3 ^d^ ± 0.03	6.8 ^d^ ± 0.14
	6	3.9 ^c^ ± 0.22	6.1 ^c^ ± 0.04	8.0 ^c^ ± 0.05	3.3 ^c^ ± 0.12	5.6 ^c^ ± 0.3	7.2 ^c^ ± 0.4
	9	4.2 ^b^ ± 0.03	6.5 ^b^ ± 0.04	8.4 ^b^ ± 0.05	3.6 ^b^ ± 0.13	6.0 ^b^ ± 0.4	7.6 ^b^ ± 0.4
	12	4.5 ^a^ ± 0.03	7.0 ^a^ ± 0.05	8.9 ^a^ ± 0.15	3.9 ^a^ ± 0.13	6.4 ^a^ ± 0.4	8.1 ^a^ ± 0.4

* The same letter in a column within a group indicates that the average values are not significantly different according to the least squares difference (LSD) test at a significance level of *p* < 0.05. ** Data presented as the mean values (±standard deviation) for twenty samples.

**Table 11 foods-14-03060-t011:** TVCs and YE count in Khlass dates with different moisture contents at +25 °C for packaging type CCC *, **.

Moisture Content	Duration	TVC	YE
(%)	(Months)	(log CFU/g)	(log CFU/g)
Control (6.235)	3	1.5 ^d^ ± 0.01	1.2 ^d^ ± 0.01
	6	1.8 ^c^ ± 0.02	1.5 ^c^ ± 0.02
	9	2.2 ^b^ ± 0.02	1.8 ^b^ ± 0.02
	12	2.5 ^a^ ± 0.03	2.2 ^a^ ± 0.03
Group A (14.877)	3	2.0 ^d^ ± 0.02	1.8 ^d^ ± 0.02
	6	2.5 ^c^ ± 0.03	2.2 ^c^ ± 0.03
	9	3.0 ^b^ ± 0.03	2.7 ^b^ ± 0.03
	12	3.5 ^a^ ± 0.04	3.2 ^a^ ± 0.04
Group B (21.781)	3	2.5 ^d^ ± 0.03	2.2 ^d^ ± 0.03
	6	3.0 ^c^ ± 0.03	2.7 ^c^ ± 0.03
	9	3.5 ^b^ ± 0.04	3.2 ^b^ ± 0.04
	12	4.0 ^a^ ± 0.04	3.7 ^a^ ± 0.04
Group C (26.807)	3	3.0 ^d^ ± 0.03	2.7 ^d^ ± 0.03
	6	3.5 ^c^ ± 0.04	3.2 ^c^ ± 0.04
	9	4.0 ^b^ ± 0.04	3.7 ^b^ ± 0.04
	12	4.5 ^a^ ± 0.04	4.2 ^a^ ± 0.04
Group D (32.062)	3	3.5 ^d^ ± 0.04	3.2 ^d^ ± 0.04
	6	4.0 ^c^ ± 0.04	3.7 ^c^ ± 0.04
	9	4.5 ^b^ ± 0.04	4.2 ^b^ ± 0.04
	12	5.0 ^a^ ± 0.05	4.7 ^a^ ± 0.05
Group E (38.697)	3	4.0 ^d^ ± 0.04	3.7 ^d^ ± 0.04
	6	4.5 ^c^ ± 0.04	4.2 ^c^ ± 0.04
	9	5.0 ^b^ ± 0.05	4.7 ^b^ ± 0.25
	12	5.5 ^a^ ± 0.05	5.2 ^a^ ± 0.25

* The same letter in a column within a group indicates that the average values are not significantly different according to the least squares difference (LSD) test at a significance level of *p* < 0.05. ** Data presented as the mean values (±standard deviation) for twenty samples.

## Data Availability

The data presented in this study are available on request from the corresponding author. The data is not publicly available due to privacy restrictions.

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
