# Peer review of "Development and Prediction of a Non-Destructive Quality Index (Qi) for Stored Date Fruits Using VIS–NIR Spectroscopy and Artificial Neural Networks"

_foods, 2025, doi:10.3390/foods14173060_

Round 1

Reviewer 1 Report (Previous Reviewer 2)

Comments and Suggestions for Authors

The article has improved significantly in response to the previous comments. The authors have addressed the key concerns, clarified important points, and strengthened the overall argument and presentation. The current version demonstrates sufficient quality and rigour to justify its acceptance for publication.

Author Response

Response to Reviewer 1 #3

Reviewer Comment

The article has improved significantly in response to the previous comments. The authors have addressed the key concerns, clarified important points, and strengthened the overall argument and presentation. The current version demonstrates sufficient quality and rigour to justify its acceptance for publication. 

Response to Reviewer

We would like to sincerely thank the reviewer for the positive and constructive feedback. We are truly grateful for the time and effort devoted to evaluating our manuscript throughout the review process. Your insightful comments and recommendations were instrumental in refining the quality, clarity, and scientific depth of the study.

We are particularly encouraged to know that the revisions have successfully addressed the initial concerns and that the current version meets the standards of quality and rigour expected for publication. We deeply appreciate your recognition of the improvements and your thoughtful engagement with our work. Your guidance has contributed significantly to enhancing the manuscript’s contribution to the field, and we are pleased to have had the opportunity to benefit from your expertise.

Thank you again for your support and your role in helping shape this work.

Reviewer 2 Report (Previous Reviewer 3)

Comments and Suggestions for Authors

The paper “Development and Prediction of a Non-Destructive Quality Index (Qi) for Stored Date Fruits Using VIS–NIR Spectroscopy and Artificial Neural Networks” is a resubmission of foods-3691645 with title “A New Quality Index of Date Fruits Utilizing VIS-NIR Spectroscopy and ANN Modeling at Different Packaging and Storage Conditions”, that I reviewed earlier. Thank you for considering my observations. As result I consider that the major problems were already resolved, therefore I have only some minor observations for this version.
Below you have my remarks:
1.    Table 1: The values represent the mean and standard deviation, not just mean. You should indicate, what “a, b, c, d, e, f” at superscript means. The position of moisture content db. (%) is incorrect. It should not be on the first column. There should be something like “class or samples”. Moisture content should be on top of the moisture content values. As a single representation over columns 2 and 3 or duplicated.
2.    Line 144 you have a strange symbol!!
3.    Eq. 6 and text! The number of samples is represented by “n” or “N”. Sometimes it is a difference. 
4.    Line 563 (and others) “327–1101 nm wavelength” and figures 6, 7, 8, 9, Please be consequent “wavelength” or “Wave Length”. And it would be useful if you can use the lambda symbol also. 
5.    Figures 14 – at the limit of observation!

Author Response

Response to Reviewer 2 #3

Point-by-point response to Comments and Suggestions for Authors:

Specific Comment

We would like to extend our sincere gratitude to the reviewer for the continued support and insightful feedback on both the original and revised versions of our manuscript. We deeply appreciate your careful evaluation of the resubmission titled “Development and Prediction of a Non-Destructive Quality Index (Qi) for Stored Date Fruits Using VIS–NIR Spectroscopy and Artificial Neural Networks.”

Your observations during the initial review were instrumental in helping us address key issues and significantly strengthen the scientific and structural integrity of the manuscript. We are encouraged to learn that you now consider the major concerns to have been successfully resolved.

We also thank you for your thoughtful minor observations on this version, which we have carefully considered and addressed in the revised manuscript. Your constructive input has played a crucial role in refining our work, and we value the opportunity to benefit from your expertise throughout the review process.

Thank you once again for your guidance and for contributing to the improvement of our research. 

General Comments

Comment 1: Table 1: The values represent the mean and standard deviation, not just mean. You should indicate, what “a, b, c, d, e, f” at superscript means. The position of moisture content db. (%) is incorrect. It should not be in the first column. There should be something like “class or samples”. Moisture content should be on top of the moisture content values. As a single representation over columns 2 and 3 or duplicated.

Response 1: We sincerely thank the reviewer for the careful and constructive feedback regarding Table 1. In response, we have thoroughly revised the table to clearly reflect the mean ± standard deviation values, as originally intended. The superscript letters (“a, b, c, d, e, f”) have now been explicitly defined in the table footnote, indicating significant differences according to the LSD test at p < 0.05.

Additionally, we acknowledge the concern regarding the placement of “Moisture content db. (%)”. We have corrected its position by introducing a more appropriate column header, “Class,” and reorganized the moisture content label to appear clearly above its corresponding data, spanning the relevant columns for clarity.

These changes have been implemented in the revised version (see Table 1, lines 126–130, highlighted in Bright Green).

We hope that these modifications enhance both the clarity and scientific rigor of the table, and we are grateful for your insightful observations. 

Comment 2: Line 144, you have a strange symbol!!

Response 2: We thank the reviewer for pointing out the unclear symbol noted in Line 166. Upon review, we confirm that the symbol in question is the internationally recognized food contact material symbol, indicating that the packaging used is safe and certified for direct contact with food products. This icon (a fork and wine glass) is standardized under EU Regulation No. 1935/2004 and other global food safety guidelines.

To improve clarity, we have now explicitly stated this in the revised manuscript as follows:

" Each package bores the internationally recognized food-contact symbol (ISO 7000-2601)." These changes are included in the revised version (see line 165, highlighted in bright green).

We appreciate the reviewer’s attention to detail, which has helped improve the clarity and completeness of the manuscript. 

Comment 3: Eq. 6 and text! The number of samples is represented by “n” or “N”. Sometimes it is a difference.

Response 3: We sincerely thank the reviewer for highlighting the inconsistency in the notation of the sample size in Equation 6 and its accompanying text. You are correct in pointing out the distinction between "n" and "N," where "N" typically denotes the total number of observations, and "n" may refer to a subset or sample.

In our case, the correct representation is "N", indicating the total number of samples used in the model. We acknowledge that earlier usage may have caused ambiguity. This has now been corrected in both the equation and the text to maintain consistency and clarity throughout the manuscript.

These revisions have been carefully implemented in the updated version (see lines 319-323, highlighted in Bright Green).

We appreciate your attention to this technical detail, which contributes to the scientific accuracy and rigor of our work. 

Comment 4: Line 563 (and others) “327–1101 nm wavelength” and figures 6, 7, 8, 9, Please be consequent “wavelength” or “Wave Length”. And it would be useful if you can use the lambda symbol also.

Response 4:

We are grateful to the reviewer for this valuable observation. Consistence in scientific terminology is essential, particularly in spectral data interpretation. In response to your suggestion, we have standardized the use of the term to “wave length” throughout the manuscript, replacing all instances of the inconsistent form “WaveLength.”

Additionally, the Greek lambda symbol (λ) has been incorporated where appropriate to enhance scientific clarity and align with standard spectroscopic notation. These improvements have been carefully implemented in the revised manuscript (see lines 606, 609, 638, 641, highlighted in Bright Green), including figures 6 through 9.

We sincerely appreciate your attention to detail which has contributed to enhancing the precision and presentation quality of the manuscript. 

Comment 5: Figures 14 – at the limit of observation!

Response 5: We sincerely appreciate the reviewer’s observation regarding the visibility and clarity of Figure 14. In response, we have carefully revised and enhanced both subfigures (A) and (B) to improve legibility. Specifically, we have increased the resolution and line thickness, adjusted the contrast of the bar colors, and standardized font sizes across all axis labels, legends, and figure captions to ensure that the information is clearly observable when printed or viewed digitally.

Furthermore, we retained all original sample identifiers and legends for scientific transparency and traceability. These visual improvements aim to support the interpretability of sensory evaluation scores across the different treatment conditions.

The revised figure has been updated in the manuscript (see Figure 14, lines 1058-1064, 1065-1071, 1072-1074, highlighted in Bright Green).

We hope that the enhancements now facilitate a more effective visual assessment of the results. 

Thanks to the reviewer for the review. We tried our best to answer/modify such comments/suggestions.

Reviewer 3 Report (New Reviewer)

Comments and Suggestions for Authors

In this contribution, Elamshit and Alhamdan, reported on the development of a non-destructive Quality Index (Qi) for stored date fruits based on Vis-NIR spectroscopy and artificial neural networks. The authors contended that the proposed system enables real-time, non-invasive quality monitoring and could support automated decision-making in postharvest management, packaging selection, and shelf-life prediction. Broadly speaking, the research is sound and based on solid evidence. The presentation is also clear and accessible, albeit quite verbose in some instances. In all, the work is of high quality, relevant, interesting, and would constitute a significant contribution to foods. However, prior to publication, the authors should address a few minor issues in the manuscript.

Line 17: TSS should be written in full given that this is the first time it is being mentioned.

-Line 76: For consistency, please correct ‘Tamer’ to ‘Tamr’.

-Figure 1 and 2 should be combined into one figure as A, B, and C.

-Line 107: ‘designed’ should be changed to ‘designated’.

-Line 134: Please check for grammar.

-Lines 746-748: Authors should add relevant references.

-Figure 12 and Figure 13 should be combined into one figure.

-Just before the Conclusions section, authors should add a paragraph delineating the limitations of the proposed approach for monitoring and evaluating quality and shelf life of date fruits.

Author Response

Response to Reviewer 3 R#3

Point-by-point response to Comments and Suggestions for Authors:

Specific Comment

We sincerely thank the reviewer for their thoughtful and encouraging evaluation of our manuscript. We are truly pleased to know that the research has been perceived as sound, relevant, and of high quality. Your recognition of the scientific merit and potential contribution of our work to the field of food quality assessment is deeply appreciated.

We also acknowledge your observation regarding the manuscript's verbosity in certain sections. In response, we have carefully revised the manuscript to improve conciseness while preserving technical clarity and completeness. Specific attention has been given to streamlining overly descriptive passages and eliminating redundancies, particularly within the Introduction and Discussion sections.

Furthermore, we have addressed all minor issues you highlighted in the manuscript with due diligence. These modifications are clearly indicated in the revised version (highlighted in bright green) and are also detailed point-by-point in the accompanying cover letter.

We remain grateful for your constructive insights, which have meaningfully contributed to enhancing both the clarity and scientific presentation of the study. We are hopeful that the revised version meets the expectations for publication in Foods and sincerely thank you for your valuable time and feedback. 

General Comments

Comment 1: Line 17: TSS should be written in full given that this is the first time it is being mentioned.

Response 1:

We sincerely appreciate the reviewer’s careful attention to clarity and terminology. In response to your observation, we have revised line 17 to provide the full term “Total Soluble Solids (TSS)” upon its first mention, thereby enhancing the manuscript’s accessibility for all readers, particularly those who may be less familiar with the abbreviation. This correction has been implemented in the revised version (see line 17, highlighted in Bright Green).

We thank you once again for your constructive feedback, which continues to contribute meaningfully to the refinement of our work. 

Comment 2: Line 76: For consistency, please correct ‘Tamer’ to ‘Tamr’.

Response 2:

We sincerely thank the reviewer for noting the inconsistency. The term “Tamer” has been corrected to the accurate and standardized form “Tamr” to maintain consistency with established terminology and improve the manuscript’s clarity. This change has been implemented in the revised version (see line 77, highlighted in Bright Green).

We appreciate your meticulous review, which continues to enhance the precision of our work. 

Comment 3: Figure 1 and 2 should be combined into one figure as A, B, and C.

Response 3: We sincerely appreciate the reviewer’s thoughtful suggestion to combine Figures 1 and 2 into a single image labeled as A, B, and C. However, after careful consideration, we respectfully propose to retain the current structure, in which these figures appear separately, for the following reasons:

  1. Figure 1 is intended to provide essential contextual information, displaying a real image of the date palm tree from which samples were harvested. This visual supports the description of the field site in the Derab region of Riyadh, Saudi Arabia, and reinforces the environmental relevance of the sampling process an important aspect for reproducibility, geographic traceability, and understanding potential environmental influences on fruit quality.
  2. Figure 2, in contrast, depicts the actual fruit samples used in the experimental analysis specifically highlighting morphological characteristics of the Sukkary and Khlass cultivars at the Tamr stage. This figure is directly tied to the experimental design and modeling of the Quality Index (Qi).
  3. Merging these two distinct visuals into one would, in our view, compromise the scientific clarity, dilute the purpose of each image, and reduce the visual effectiveness in supporting their respective sections in the manuscript (sampling context vs. experimental material).
  4. Additionally, we would like to kindly note that both Reviewer 1 and Reviewer 2 expressed no concerns regarding the current figure arrangement, and their feedback implicitly supported the current layout and progression. We are concerned that altering the structure at this stage could potentially introduce conflicting opinions among reviewers and result in the figures being returned for further evaluation, which may delay the acceptance process.

We hope this explanation justifies our decision to preserve Figures 1 and 2 as separate, complementary elements, each serving a distinct scientific purpose.

We remain fully open to further guidance from the editor should a unified editorial position on this matter be required. 

Comment 4: Line 107: ‘designed’ should be changed to ‘designated’.

Response 4: We thank the reviewer for the careful reading and the attention to precise word usage. The term “designed” has been appropriately revised to “designated” in line 108, as suggested, to ensure linguistic accuracy and improve the clarity of the sentence. This change is now reflected in the revised manuscript (highlighted in Bright Green).

We appreciate your continued guidance in enhancing the quality and readability of the manuscript. 

Comment 5: Line 134: Please check for grammar.

Response 5: We sincerely appreciate the reviewer’s attention to grammatical clarity and structure. In response, we have revised the paragraph on packaging materials to enhance their grammatical correctness and improve readability. The revised version now clearly lists the five packaging types and provides a smoother flow of information. These changes have been implemented in the updated manuscript (see lines 138–141, highlighted in Bright Green). We are grateful for your suggestion, which has helped us improve the presentation and precision of the methodology section. 

Comment 6: Lines 746-748: Authors should add relevant references.

Response 6: We sincerely thank the reviewer for highlighting the importance of supporting our claims with relevant references. In response, we have carefully revised the specified section (lines 790–795, highlighted in Bright Green) by incorporating five authoritative and up-to-date references that reinforce the discussion and enhance the scientific credibility of our findings. These references were selected to align closely with the context of model validation and performance indicators (RMSEP, REP, and RER) in VIS–NIR and ANN-based quality prediction models.

We believe these additions strengthen the manuscript's rigor and provide readers with a clearer understanding of the robustness and relevance of our modelling approach. 

Comment 7: Figure 12 and Figure 13 should be combined into one figure.

Response 7: We sincerely thank the reviewer for this thoughtful suggestion. While we fully understand the intent to streamline the visual presentation by combining Figures 12 and 13, we respectfully believe that maintaining them as separate figures ensures greater clarity and scientific precision. The two figures illustrate distinct analytical dimensions:

  1. Figure 12 presents the effect of packaging type over storage time at a fixed temperature (–18 °C) on hardness, highlighting material-dependent mechanical stability.
  2. Figure 13, on the other hand, evaluates the combined influence of storage time and temperature on moisture content, using a fixed packaging material (PSSPC), and is plotted in a 3D surface format to capture temperature interactions.
  3. Merging them would necessitate substantial restructuring of both axes and formats, potentially introducing graphical distortion and loss of interpretability, especially as Figure 12 uses a 2D format and Figure 13 employs a 3D topographical approach. This would also compromise the visual clarity that is critical for conveying nuanced variations in storage response between cultivars and treatment conditions.
  4. Moreover, we would like to respectfully note that both the first and second reviewers have previously expressed satisfaction with the current figure arrangement and flow. Altering the structure at this stage may introduce inconsistencies or risk conflict between reviewer perspectives, potentially prompting unnecessary resubmission rounds.

Therefore, we kindly request retaining the current figure layout to preserve both scientific clarity and reviewer consensus. We are, of course, open to additional editorial guidance if deemed necessary. 

Comment 8: Just before the Conclusions section, authors should add a paragraph delineating the limitations of the proposed approach for monitoring and evaluating quality and shelf life of date fruits.

Response 8: We sincerely thank the reviewer for this valuable suggestion, which we believe adds an important dimension to the scientific transparency and practical relevance of our study. In response, we have added a dedicated paragraph just before the Conclusions section (see line 1113-1126, highlighted in Bright Green) that thoughtfully outlines the limitations of the proposed non-destructive Quality Index (Qi) framework.

This addition critically addresses factors such as the dependency on calibration consistency, potential spectral noise under field conditions, and the need for cultivar-specific model tuning. We believe this clarification not only enhances the manuscript's rigor but also provides a balanced view of the approach’s applicability and scope for future refinement. We truly appreciate the reviewer’s insight in prompting this essential improvement. 

Thanks to the reviewer for the review. We tried our best to answer/modify such comments/suggestions.

This manuscript is a resubmission of an earlier submission. The following is a list of the peer review reports and author responses from that submission.

Round 1

Reviewer 1 Report

Comments and Suggestions for Authors

Comments:

Abstract

  • Line 18: Authors should report the numerical result in the abstract.

Introduction

  • Introduction need to be shortened.

Materials, Methods, Measurements

  • Figure 1 a not clear. Authors need it to make clearer.
  • The authors should provide a clear justification for the use of -18°C. Even in the absence of experimental data, it is evident that -18°C could be considered optimal; however, this assumption still requires proper scientific reasoning.
  • Table 1 Moisture should be report in the form of the Mean ± SD.
  • Proper caption needs to be considered for the Table 1.
  • Line 240: “0.225 mL” should be revised. Authors need to recheck this amount.
  • Quality Index (Qi) Prediction need to be rewrite in clearer way.

Results and Discussions

  • Overall results and discussion need to be improved. Authors need to compare their result with other recent researcher.

Author need to justify the reason on confident interval during the storage why microbial is each 3 month and other experiment only 2 months. This type of comparison not align. 

Reviewer 2 Report

Comments and Suggestions for Authors

The manuscript presents a well-structured experimental study on the application of VIS-NIR spectroscopy combined with Artificial Neural Networks (ANN) and Partial Least Squares Regression (PLSR) for non-destructive quality assessment of date fruits under varying storage and packaging conditions. The research is timely, especially for postharvest quality monitoring and food safety management in high-value crops like dates. However, despite the comprehensive approach and large dataset, there are several concerns that warrant attention to improve the rigor, transparency, and novelty of the manuscript.

General Comments

The introduction could be significantly shortened and made more concise. The current version includes excessive general background, and it lacks focus on the objective and novelty of the manuscript.

The manuscript refers to the maturity stages of date fruits without explanation. Please describe these stages (Hababouk, Kimri, Khalal, Rutab, Tamar) briefly to aid understanding for non-specialist readers.

Lines 69–71: The statement in these lines lacks relevance and does not add meaningful information. Consider removing or replacing it with content that better supports the study’s rationale.

The novelty of the proposed Quality Index (Qi) is not clearly articulated. The manuscript cites three previous works using similar approaches but does not clarify what differentiates this Qi or how it advances the field. This should be explicitly addressed.

More details are needed about the packaging types. Were they food-grade certified? Who was the supplier or manufacturer? This information is important for reproducibility and safety validation.

The procedure to artificially moisturize dates with six dosages of water lacks scientific validation. There is no evidence that this method leads to uniform moisture distribution or replicates the physiological conditions of naturally high-moisture dates. Is this practice standard in the industry? Please explain its relevance and validity.

Grouping samples solely by moisture content is insufficient without addressing other confounding variables such as sugar crystallization or fiber content. Provide average values and variability of these parameters within groups to support the grouping strategy.

In Formula 5, specify whether the maximum and minimum values used for normalization are calculated per day or for the entire dataset.

Clarify how the 15 000 spectral samples were split into training, testing, and validation sets. Were instrumental replicates considered, and if so, were they kept grouped or randomized? This has implications for model independence and performance.

The ANN model architecture is not disclosed. Please provide detailed information on the number of layers, activation functions, training epochs, learning rate, regularization methods, and software used. This is essential for reproducibility.

The explanation that “LVs were determined by exploring sub-spaces that enhance covariance between the predictor and response variables” is vague. Since PLSR maximizes the covariance between X and Y, restate this clearly and accurately.

RMSEC and RMSECV are not sufficient alone to evaluate model performance. Include additional metrics such as Relative Error of Prediction (REP) and Range Error Ratio (RER), along with their formulas. Also, for the validation set of 1500 samples, report the RMSEP values.

The use of a second-order polynomial for modeling Qi should be justified. A first-order (linear) model could potentially provide similar accuracy with better interpretability and reduced risk of overfitting. Consider comparing both fits or at least justifying the second-order choice.

(see figure attached)

Figure 1 is merely illustrative and does not contribute substantively to the scientific content. It should be moved to the supplementary materials.

Please include detailed information on packaging types directly in the captions of all relevant figures to facilitate reader understanding.

Include error bars in Figure 13 to represent the variability in sensory evaluation scores.

While microbial analysis is highlighted as important, detailed results are not shown in figures or tables. Please provide full CFU count data and relevant statistical analysis.

Ethical approval for sensory testing is not appropriately addressed. Given that human participants were involved in the sensory evaluation of stored date fruits, the study qualifies as human subjects research. The manuscript should clearly state that ethical approval was obtained from an Institutional Review Board (IRB) or equivalent ethics committee. Additionally, it should confirm that informed consent was obtained from all participants, in compliance with ethical standards for human research.

Reviewer 3 Report

Comments and Suggestions for Authors

The paper “A New Quality Index of Date Fruits Utilizing VIS-NIR Spectroscopy and ANN Modeling at Different Packaging and Storage Conditions” was an interesting paper to be read. This aim to evaluate the quality of Sukkary and Khlass dates fruits by employing a series of evaluation parameters for samples stored in different conditions of temperature (-18 C, 5 C and 25 C) and packed using various packaging materials (OCC, CCC, CCSPB, SSPC and PSSPC). With a major exception, the manuscript is well written, and I confess that I liked the characterization of stored date fruits (SDF) by physicochemical properties, the assessment of microbial quality and the sensory evaluation. The exception is related to the main theme of the paper, e.g. the newly introduced dimensionless and normalized Quality Index (Qi) and the prediction of this, the VIS-NIR spectroscopy and spectra and the ANN. In these aspects your manuscript, in my opinion, failed to provide enough details for a reader to understand the procedures (see below). Nevertheless, recognizing the importance of reported data and your laborious work, my recommendation to the editor was that the paper to be, this time rejected and rewritten (see my numerous observations) and then resubmitted for publications.Below you have my remarks:1.    Abstract: “Qi was predicted using partial least squares regression (PLSR) and artificial neural networks (ANN) based on VIS-NIR spectral data” – the readers would like to know from abstract what have you measured exactly? Can you be more specific and define the state of samples?2.    Abstract. Please define the terms “PSSPC”, “OCC” and “CCC”. 3.    Lines 43-44 “Dates experience substantial losses throughout the market chain, amounting to up to 26%.” – Please specify clearly what is lost! Total mass or just water/moisture?4.    Lines 61-62 “the reflectance or absorbance of measured spectrophotometric waves and the physicochemical properties of food products” – the reflectance or absorbance is a property of a material, the electromagnetic wave are reflected or absorbed. Please rephrase!5.    Lines 65-66 “the interaction of near-infrared light with sample molecules” – if it is light it is not in near-infrared. You may write, for example “near-infrared electromagnetic wave” or “near-infrared radiation”. The first one is better.6.    Lines 66-67 “leading to specific light absorption, transmission, or reflection patterns” – if it is a pattern the usually some phenomena like, interference, diffraction are involved. Here I don’t think that it is the case! Please rephrase!7.    Lines 67-68 “By measuring light intensity at different wavelengths, artificial neural networks (ANNs) can be utilized in NIR analysis” – Measuring the light intensity function of the wavelength means to record a spectrum, then to make spectroscopy. You may “feed” (provide the spectrum as initial data) for an ANN. Please describe appropriate these two operations. 8.    Line 71 “were reviewed [17].” – be more specific. Where? Or use an adjective!9.    Line 75 “based on Vis-NIR can classify various fruits using wavelengths of 300-1041 nm” – you present a range which go a little also in near UV domain. Please be more specific!10. Line 95 – please define the acronym “HPLC”.11. Lines 91-104 – “NIR” is just the range. Here you have to specify the method (spectroscopy) in the context of: “was shown”, “was utilized”, “helped”, etc. Please take into account this aspect, here and everywhere!12. Line 104 “was provided [28]” – please be specific! Where? In ref. [28]? Then say so!13. Line 112 “[31]The” – dot and space is missing!14. Line 146 “They were cleaned with air” – please be more specific. How can air clean? Air jet under pressure?15. Line 146 “sorted from damaged and non-consumable fruits”. Please be more specific. You introduced only two categories, and I guess that these are not the used ones!16. Figure 2. The resolution is so low then I can’t say that these are the “ready-to-measure fruits”. Please increase resolution.17. Line 154 what do you mean by “six dosages of distilled water”?18. Lines 156-157: There is a contradiction between “for a predetermined period” and “Once equilibrium was reached … date samples were placed”. Which one is correct?19. Line 160 “All examined samples were scanned with Felix 950 for NIR assessment” – is totally unclear. What kind of device is “Felix 950”? Function of that the second part “NIR assessment” should be adapted! Moreover you claim, in title, to use “VIS-NIR Spectroscopy”!20. Section “c) Chemical analysis”. The reader it is interested not by a declaration but to enumerate the types of analysis!21. First paragraph from “b) VIS-NIR Technique”. Is the “F-750” the same with Felix 950? 22. Line 282 “VIS-NIR spectrometer operating in the range 285–1200 nm” it is a UV-VIS-NIR spectrometer! This is the range of spectrometer. Which range you used? (see the line 286 “Pre-processing NIR spectrum data”) should be clearly stated!23. Line 287 “Utilizing derivatives can be particularly beneficial in (NIR) spectroscopy” – please specify what kind of “derivatives”! Numerical on a function? Food derivatives?24. Line 289 “to ensure complete contact with the date surface of the lens of the F-750” you should clearly specify what kind of spectroscopic method you use (Reflectance, transmission/absorption?), and the state of samples. Describe better the experimental arrangement light source, sample, holder, lens, dispersive element (refraction grid, prism?), and acquisition sensor! If possible a figure will be more suggestive. Ah, in this sense lines 295-305 are better to be placed with one paragraph before!25. Line 306 “After capturing the spectrum” – usually an image is captured, while a spectrum is recorded!26. Line 308 usually “Savitzky–Golay” is known as a smoothing procedure (and a second order polynomial can be used). Did you do something else? A second derivative? For what purpose? Please explain!27. Line 312 “artificial neural network (ANN), were utilized to develop calibration models”. Usually, when an ANN is trained, the inputs are: data + label (or value) and is set for classification or regression. Please specify these in your case (classes)! Your ANN will predict then what?28. Lines 313-316 please provide an example of latent variables (LVs) for your problem!29. Figure 4. The vertical axis can start from 0.6, there is a large unused blank space. Fig. 4 legend: Please clarify the term “predicted”. How it was predicted!?30. Table 2 and the mathematical procedure. It is known from many centuries that this procedure is called fitting and then interpolating with a second order polynomial. It is a rigorous procedure. The used term “prediction” assumes a less clear result. Please use the established term! “Prediction” attribute can be used not for interpolated data (as presented in Figure 4) but for some extrapolation beyond 12 months. But the second order polynomials are inappropriate for these long term predictions. For example I performed such experiments (predictions) and I found that for Sukkary the Qi index will be negative in less than 31 months. For Khlass, it is worse: In 12 months and few days the Qi (assumed as second order polynomials) will have a minimum, and then Qi index will increase, reaching Qi = 1 (perfect dates) symmetrically in less than 25 months, and will increase further. Not all the functions that may fit some data can have a physical meaning. My advice for you is to rewrite entirely this part based on other functions like exponential, Gaussian or better a kind of sigmoidal function.31. Lines 370-372 “It can be seen that the quality index is a strong and reliable indicator for describing and assessing the overall quality of fruits during storage under different conditions of storage temperature and packaging for date cultivars.” – In the light of the previous comments for the moment I can see that!32. Line 375 “This also indicates the possibility of predicting the actual shelf life of other fruits and vegetables.” This statement was just infirmed. 33. Line 382: “The mean second absorbance derivative spectra” - did you mean “The mean second derivative of absorbance spectra”? With which purpose? Please explain the new information obtained from second derivative compared to the spectra! (and compared with the first derivative), and the noise handling!34. Line 383 “spanning from 304.5 to 1123.5 nm wavelength range” – finally the range!!! You should provide the information also earlier. And, you have to correct in all places where you discuss about your measurement and NIR alone appear. It is VIS-NIR or even NUV-VIS-NIR spectroscopy!!!35. Lines 386 “Figure 5. shows an image of the raw spectra absorbance” and 394 “Figures 6 and 7 show the reflectance spectra (%)”. Please clarify the measurement and analysis procedure: reflectance or absorbance? Or both but explain!36. Section “3.3.1 Partial Least Squares Regression (PLSR)” you describe with many details the performance of the method but the effective results are not presented at all. What is your goal? What are you trying to do? You did not explain which features of VIS-NIR spectra you use and correlate with what? Qi? Measured monthly? Please describe the calibration analysis. Usually imply that a parameter 1 is plotted (correlated) function of parameter 2. Which are these parameters in your case? In this sense lines from 430 to 443 (including Table 3) and from 456 to 481 (including Table 4) are less relevant since the object is missing. 37. Line 486 “under the conditions of this study, it is recommended to use the ANN” – this is what is missing from your paper. A basic description of how it is used! Just guessing about how it is used; a graph of actual Qi and the predicted Qi by ANN will be more informative. 38. Lines 612-614 and Figure 12 “At -18°C, the moisture content remains almost constant over 12 months, demonstrating the effectiveness of low temperatures in maintaining moisture content through storage.” – True! but after a significant initial drop! Please discuss this aspect. 39. Figures 12 and 13. I can see a legend of 21 samples and graphical representation of 9 samples. Please explain. In figure legend is missing the information related time of storage in the moment of sensory preference.

Round 2

Reviewer 2 Report

Comments and Suggestions for Authors

Authors have answer many of my previous question, improvind the scientific quality of the manuscript. However, some responses do not meet the required changes in the manuscript:

The current justification for omitting additional performance metrics is not convincing. While RMSEC and RMSECV are commonly used, they are not sufficient alone to assess model robustness and predictive accuracy—especially in chemometric modelling, where multiple Figures of Merit (FoM) are standard. REP and RER are easy to compute, provide complementary information, and are frequently recommended in the literature to evaluate predictive error relative to signal range and magnitude. Additionally, since you have an independent validation set of 1500 samples, reporting RMSEP is both expected and valuable. These metrics should be included in the current manuscript, not deferred to future work, to strengthen the credibility and completeness of your model assessment.

Although second-order models yielded marginally higher R² values, the improvement over first-order models is relatively small (e.g., from 0.976 to 0.988). This does not clearly justify the increased model complexity, especially without additional statistical tests (e.g., ANOVA, F-test, or information criteria like AIC) to determine if the difference is significant. In the absence of such evidence, the simpler first-order model may be preferable in line with parsimony principles. Please consider either reverting to the linear model or providing formal justification that the second-order model adds meaningful explanatory power.

While the choice of a radar chart helps visualize sensory profiles, the lack of error bars makes it impossible to assess whether observed differences are statistically meaningful or within the range of variability. Even if radar plots cannot accommodate error bars clearly, alternative visualizations (e.g., bar plots with error bars or individual attribute plots) should be used to support the interpretation of sensory differences. I strongly recommend including a supplementary figure or redesigning the figure to incorporate error margins in a scientifically rigorous way.

Reviewer 3 Report

Comments and Suggestions for Authors

Thank you for considering to respond positively to all my suggestions. 

Author Response

ROUND 2 REVIEWER 3:

Comment 1: Thank you for considering to respond positively to all my suggestions. 

Response 1: Thank you so much for your review and the very positive response to all our modifications in Round 1.

Best wishes for all..